# QUADGPT: NATIVE QUADRILATERAL MESH GENERATION WITH AUTOREGRESSIVE MODELS

**Jian Liu**[1,2]    **Chunshi Wang**[2]    **Song Guo**[1*]    **Haohan Weng**[2]    **Zhen Zhou**[2]    **Zhiqi Li**[2]
**Jiaao Yu**[2]    **Yiling Zhu**[2]    **Jing Xu**[2]    **Biwen Lei**[2]    **Zhuo Chen**[2]    **Chunchao Guo**[2*]
[1]Hong Kong University of Science and Technology.    [2]Tencent Hunyuan.

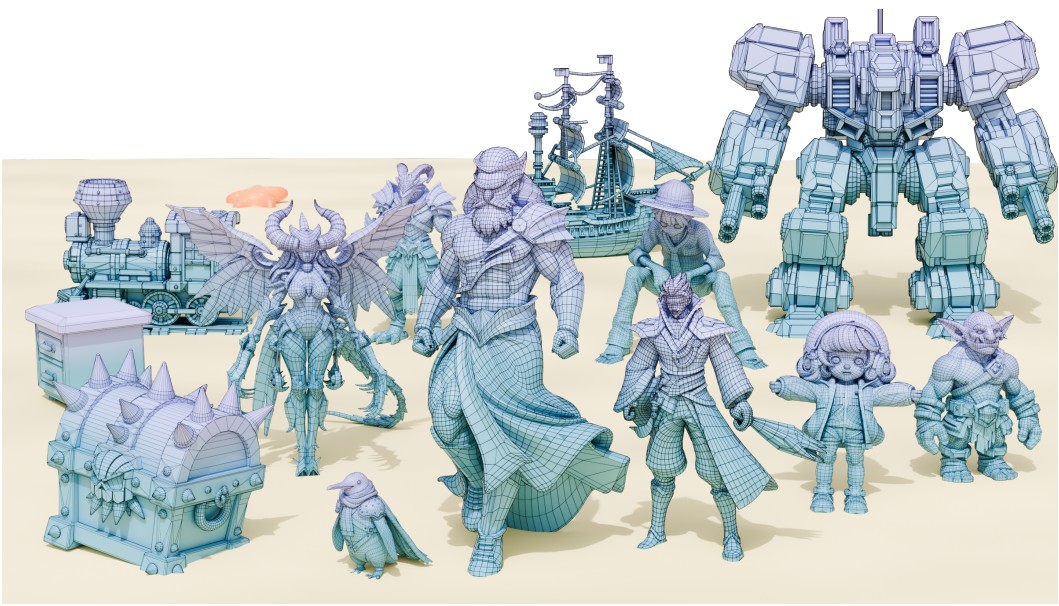

Figure 1: **QuadGPT** can generate diverse, high-quality quad meshes conditioned on point clouds.

## ABSTRACT

The generation of quadrilateral-dominant meshes is a cornerstone of professional 3D content creation. However, existing generative models generate quad meshes by first generating triangle meshes and then merging triangles into quadrilaterals with some specific rules, which typically produces quad meshes with poor topology. In this paper, we introduce QuadGPT, the first autoregressive framework for generating quadrilateral meshes in an end-to-end manner. QuadGPT formulates this as a sequence prediction paradigm, distinguished by two key innovations: a unified tokenization method to handle mixed topologies of triangles and quadrilaterals, and a specialized Reinforcement Learning fine-tuning method tDPO for better generation quality. Extensive experiments demonstrate that QuadGPT significantly surpasses previous triangle-to-quad conversion pipelines in both geometric accuracy and topological quality. Our work establishes a new benchmark for native quad-mesh generation and showcases the power of combining large-scale autoregressive models with topology-aware RL refinement for creating structured 3D assets.

## 1    INTRODUCTION

In the creation of high-fidelity 3D assets for game development, quadrilateral-dominant meshes play a fundamental role in ensuring modeling efficiency, deformation stability, and animation readiness. The structured topology provided by quad-dominant layouts facilitates smoother subdivision surfaces, more natural articulation, and easier UV unwrapping, all of which are essential in production

---

*Corresponding authors.

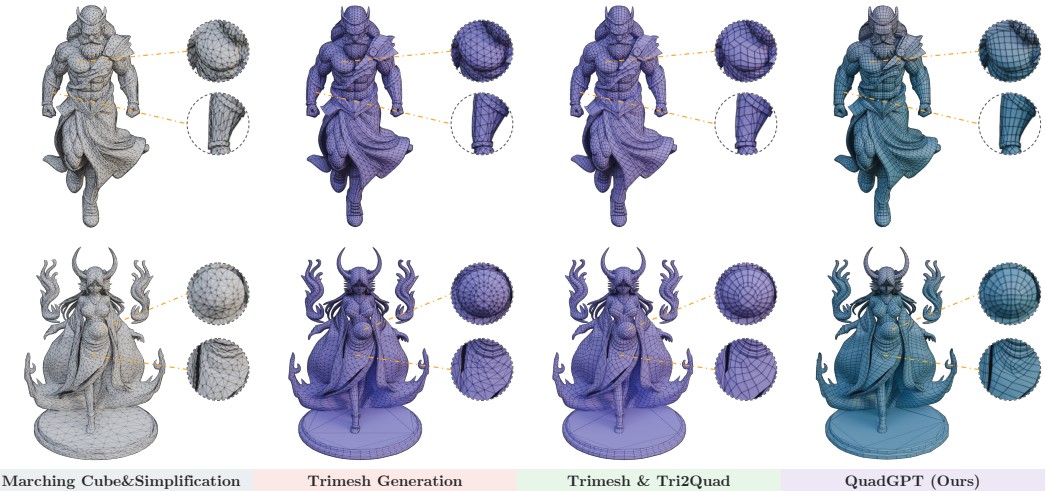

| Marching Cube&Simplification | Trimesh Generation | Trimesh & Tri2Quad | QuadGPT (Ours) |

Figure 2: **Comparison of topological quality across different pipelines.** Iso-surfacing methods produce dense, unstructured triangular meshes. Autoregressive triangle generation followed by heuristic conversion fails to create coherent structure. Our QuadGPT directly generates native quadrilateral meshes with clean, artist-friendly edge flow.

environments (Lei et al., 2025). Consequently, quad meshes have become the industry standard, especially in character and organic modeling, where clean edge flow and controllable curvature are critical.

In the pursuit of automated generation of artist-ready 3D assets from inputs such as text or images, existing approaches typically decouple the problem into two separate stages: geometry generation and topology generation. For geometry, latent diffusion models (LDMs) have achieved remarkable success in generating 3D shapes (Zhao et al., 2025b; Li et al., 2025d) via implicit representations like SDF (Li et al., 2024). Meshes are then extracted with iso-surface algorithms (Lorensen & Cline, 1998; Shen et al., 2021; 2023), inevitably resulting in unstructured dense meshes. On the other hand, topology-focused methods using cross-field guidance (Huang et al., 2018; Dong et al., 2025a) aim for structure but are often not robust, requiring pristine input meshes and failing to produce the adaptive, artist-like tessellation where polygon density matches geometric complexity.

More recently, autoregressive mesh generation methods such as MeshAnything (Chen et al., 2025b), BPT (Weng et al., 2025) and Mesh-RFT (Liu et al., 2025) have shown promise by modeling mesh sequences with Transformer architectures. While these approaches capture artist-like topology, they remain limited to generating only triangular meshes. Converting these outputs into quadrilateral meshes still requires triangle-merging algorithms that often break natural edge flow and introduce artifacts, as demonstrated in Figure 2. Consequently, even high-quality triangle meshes are hard to translate into production-ready quad layouts, highlighting a fundamental discrepancy between generated 3D assets and industrial applications.

To address these challenges, we propose an end-to-end autoregressive framework for direct generation of native quadrilateral meshes. Our model consumes a point cloud as input and produces a structured face sequence as output. Recognizing that artist-crafted meshes are typically quad-dominant yet usually incorporate a small number of triangles, we design a novel unified representation that explicitly supports mixed-element topologies through a tailored padding strategy for triangular faces. For computation efficiency, we employ an Hourglass Transformer architecture that first condenses the face sequence and subsequently compresses vertex information. The model is trained using a truncated sequence strategy, enabling support for high-poly meshes. To further improve topology quality, we introduce a reinforcement learning fine-tuning phase with truncated direct preference optimization (tDPO) that rewards the formation of coherent edge loops, a characteristic feature of professionally designed assets. Our reinforcement learning framework is specifically designed to evaluate and compare truncated sequences, ensuring effective optimization even for large-scale meshes.

Extensive experiments on the Toys4K dataset (Stojanov et al., 2021) confirm that QuadGPT consistently generates higher-quality 3D meshes than state-of-the-art baselines. By leveraging a large-scale, carefully curated dataset and comprehensive pre-training and post-training protocols, we further evaluate QuadGPT on dense meshes generated by Hunyuan3D (Zhao et al., 2025b). The model demonstrates robust performance across both soft-surface models (e.g., human characters) and hard-surface objects (e.g., props). To ensure a rigorous comparison, we trained an triangle-only variant (TriGPT) followed by triangle-to-quad conversion. As shown in Figure 2, QuadGPT's native quadrilateral architecture yields significantly superior topology. This breakthrough establishes QuadGPT as an unequivocally state-of-the-art solution, effectively bridging the gap between text/image inputs and production-ready 3D artist meshes.

The main contributions of this paper are summarized as follows:

- We present QuadGPT, the first autoregressive model that generates native quad-dominant meshes in an end-to-end manner.

- We propose a unified sequence representation for mixed-element meshes with a padding-based serialization, enabling the scalable process of heterogeneous mesh topologies.

- We introduce tDPO, which is designed to optimize global quadrilateral flow through a novel reward mechanism that encourages the formation of structured edge loops.

- QuadGPT achieves state-of-the-art performance, producing game-ready meshes that exceed existing methods in both geometric and topological quality.

## 2 RELATED WORK

**Indirect and Field-Guided Mesh Generation.** A dominant paradigm in 3D generation relies on continuous neural representations like vecsets (Zhang et al., 2023; 2024; Li et al., 2024; Zhao et al., 2023; Wu et al., 2024; Li et al., 2025b; Zhao et al., 2025b) or sparse voxels (Xiang et al., 2024; Ye et al., 2025; He et al., 2025; Wu et al., 2025; Li et al., 2025d). A universal limitation of these methods is their reliance on an iso-surfacing step, such as Marching Cubes (Lorensen & Cline, 1998), which invariably yields dense, topologically unstructured **triangular meshes**. Concurrently, traditional approaches to quadrilateral meshing are predominantly guided by cross-field computation. These methods are either optimization-based, requiring slow, per-shape optimization (Bommes et al., 2009; Jakob et al., 2015; Huang et al., 2018; Bommes et al., 2013; Ebke et al., 2016; Campen & Kobbelt, 2014; Jiang et al., 2015; Diamanti et al., 2015), or more recent learning-based techniques that accelerate field prediction (Dielen et al., 2021; Li et al., 2025c; Dong et al., 2025b;a). However, all field-guided methods depend on multi-stage pipelines that are not end-to-end generative frameworks.

**Native Triangle Mesh Generation.** To address the limitations of indirect methods, a promising direction has been the direct autoregressive generation of mesh sequences. This field, pioneered by MeshGPT (Siddiqui et al., 2024), has seen rapid progress. Subsequent work has largely focused on three key areas: (1) developing more efficient tokenization and compression schemes to manage long sequences (Chen et al., 2024; 2025c; Tang et al., 2025; Weng et al., 2025; Lionar et al., 2025; Song et al., 2025; Kim et al., 2025); (2) achieving scalability to tens of thousands of faces through architectural innovations, most notably with the Hourglass Transformer in Meshtron (Hao et al., 2024); and (3) enhancing output quality and alignment with human preferences via reinforcement learning, as demonstrated by DeepMesh (Zhao et al., 2025a) and Mesh-RFT (Liu et al., 2025). Complementary efforts have focused on areas such as acceleration (Chen et al., 2025a; Wang et al., 2025a;b) and continuous level-of-detail (Zhang et al., 2025). Despite these significant advances, all existing methods in this domain are fundamentally confined to generating **triangular meshes**. This reveals a critical gap between the state-of-the-art in generative modeling and the practical need for industry-standard quadrilateral assets. Unlike the above works, QuadGPT makes the first attempt to bridge this gap, presenting a scalable generative model for native quadrilateral mesh generation.

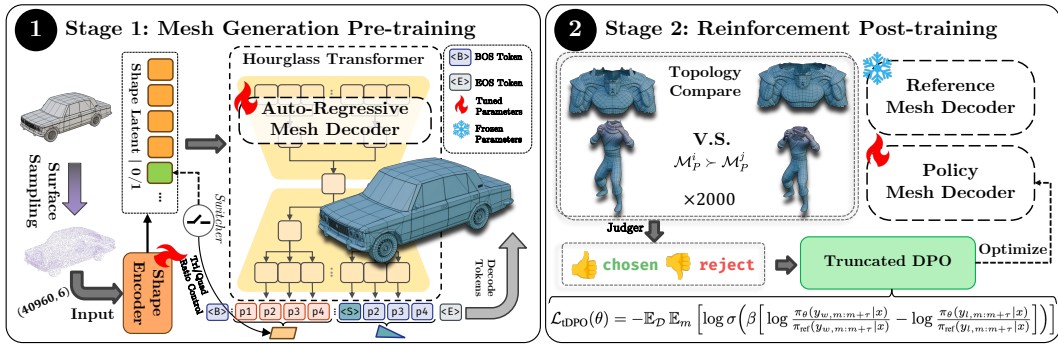

Figure 3: **Pipeline of QuadGPT.** First, an autoregressive Hourglass Transformer is pre-trained to generate mesh sequences conditioned on an input point cloud. Subsequently, the model is fine-tuned using Truncated Direct Preference Optimization (tDPO), where a preference dataset is automatically constructed by comparing truncated sequences via a novel topological reward.

# 3 QUADGPT

Our approach, QuadGPT, introduces the first autoregressive framework for the direct generation of native quadrilateral and mixed-element meshes. The methodology consists of three core pillars: (1) a unified serialization scheme to represent mixed-topology meshes as a single token sequence; (2) a powerful autoregressive architecture for generative pre-training; and (3) tDPO stage with a novel topological reward for direct topological optimization. An overview of our pipeline is presented in Figure 3.

## 3.1 UNIFIED SERIALIZATION FOR MIXED-ELEMENT MESHES

We formulate mixed-element mesh generation as a sequence prediction problem, transforming a mesh $\mathcal{M}$ into a single, linear sequence of discrete integer tokens. Our serialization scheme is designed to be canonical and to uniformly handle both triangular ($n = 3$) and quadrilateral ($n = 4$) faces. A mesh is represented at three hierarchical levels:

$$
\begin{aligned}
\mathcal{M} &= \{\mathbf{f}^1,\ \mathbf{f}^2,\ ...\ \mathbf{f}^{N_f}\} && \texttt{Face Level} \\
&= \{\mathcal{S}(\mathbf{f}^1),\ \mathcal{S}(\mathbf{f}^2),\ ...\ \mathcal{S}(\mathbf{f}^{N_f})\} && \texttt{Token Block Level} \\
&= \{\underbrace{\tau_{\text{pad}}, ..., \mathbf{c}_1^1, ..., \mathbf{c}_3^1}_{\text{e.g., Triangle Block}},\ \underbrace{\mathbf{c}_1^2, ..., \mathbf{c}_4^2}_{\text{e.g., Quad Block}},\ ...\} && \texttt{Coordinate} - \texttt{Token Level}
\end{aligned}
\tag{1}
$$

where $\mathcal{S}(\cdot)$ is the face serialization function and $\tau_{\text{pad}}$ is a special padding token.

**Canonical Representation.** To ensure a deterministic sequence for any given geometry, we first establish a canonical representation. Vertex coordinates are normalized to a $[-0.95, 0.95]^3$ cube and then quantized. To minimize precision loss, we employ a high-resolution **1024-level (10-bit)** quantization, mapping each coordinate to an integer in $\{0, 1, \ldots, 1023\}$. All unique vertices are then sorted lexicographically by $(z, x, y)$ coordinates, and the face set is sorted accordingly. This multi-step process guarantees a unique mapping from mesh to sequence.

**Unified Token Block Structure.** The cornerstone of our approach is a **unified fixed-length block representation**. Each face, regardless of its valence, is tokenized into a consistent 12-token block using the padding token $\tau_{\text{pad}}$ (integer value 1024). A quadrilateral face is tokenized by directly flattening its $4 \times 3 = 12$ vertex coordinate tokens. A triangular face is prepended with three $\tau_{\text{pad}}$ tokens, followed by its $3 \times 3 = 9$ coordinate tokens, also forming a 12-token block. This design allows the model to implicitly learn the face type from the presence of padding.

This unified structure offers several advantages: it creates a simple, highly parallelizable tokenization process that scales effortlessly; it simplifies the model architecture; and it allows the Transformer to naturally differentiate face types without explicit type tokens.

## 3.2 AUTOREGRESSIVE PRE-TRAINING

The pre-training stage is designed to teach QuadGPT the fundamental distribution of mesh geometry and connectivity by training it to predict the next token in a sequence. The model is optimized using a standard cross-entropy loss objective:

$$\mathcal{L}_{ce} = \text{CrossEntropy}(\hat{\mathbf{S}}[:-1], \mathbf{S}[1:]), \tag{2}$$

where $\mathbf{S}$ is the ground-truth token sequence and $\hat{\mathbf{S}}$ represents the predicted logits. Our approach incorporates a powerful hierarchical architecture, shape conditioning, and a specialized training strategy tailored for structured geometric data.

**Hierarchical Model Architecture.** Rather than treating mesh generation as a generic sequence task, we leverage the inherent hierarchical structure of mesh data. To this end, we utilize the **Hourglass Transformer** architecture (Hao et al., 2024; Nawrot et al., 2021), which processes the input sequence at multiple levels of abstraction. Let the input token sequence embeddings be $\mathbf{E}^{(0)} \in \mathbb{R}^{L \times D_0}$. The architecture employs a series of causality-preserving shortening layers to create a computational bottleneck. The sequence is first processed by a Transformer block and then shortened by a factor of 3, and subsequently by a factor of 4:

$$\mathbf{E}^{(1)} = \text{Shorten}_3(\text{TransformerBlock}_1(\mathbf{E}^{(0)})) \in \mathbb{R}^{(L/3) \times D_1} \tag{3}$$

$$\mathbf{E}^{(2)} = \text{Shorten}_4(\text{TransformerBlock}_2(\mathbf{E}^{(1)})) \in \mathbb{R}^{(L/12) \times D_2} \tag{4}$$

This hierarchical processing enables the model to efficiently capture high-level global context in its bottleneck layers and fine-grained local details in its outer layers before upsampling back to the original sequence length for prediction.

**Shape and Topological Conditioning.** QuadGPT's generation is guided by two primary conditions. First, a point cloud with normals, $\mathcal{P} = \{\mathbf{p}_i \in \mathbb{R}^6\}_{i=1}^{N_p}$, is encoded into a global shape embedding $\mathbf{E}_{\text{shape}}$ by a pre-trained Michelangelo encoder (Zhao et al., 2023). To ensure this geometric context remains influential throughout the generation of long sequences, the embedding is supplied to the decoder via cross-attention:

$$\mathbf{H}' = \text{CrossAttn}(\mathbf{H}, \mathbf{E}_{\text{shape}}, \mathbf{E}_{\text{shape}}), \tag{5}$$

where $\mathbf{H}$ represents the decoder's hidden states. Second, to enable our training curriculum, we introduce a learnable embedding conditioned on a quad-dominance parameter $r \in [0, 1]$. This parameter explicitly controls the target ratio of face types, from purely triangular ($r = 0$) to mixed quadrilateral ($r = 1$), providing the mechanism for our curriculum learning strategy.

**Training Strategy.** Our training strategy combines truncated sequence training for efficiency with a novel curriculum for stability. To manage the long sequences of high-resolution meshes, we employ **truncated training**, using fixed-length segments (e.g., 36,864 tokens) to enable efficient, large-batch processing. Furthermore, to ensure stable learning of complex quadrilateral topology, we introduce a **curriculum learning** strategy. We first initialize QuadGPT with weights from a model pre-trained exclusively on triangular meshes. We then progressively finetune this model, using our quad-dominance condition $r$ to gradually anneal the training data distribution from purely triangular ($r = 0$) to quad-dominant ($r \to 1$). This graduated exposure allows the model to master basic geometric syntax before tackling the more complex rules of quadrilateral topology, significantly improving stability and convergence speed.

**Data Strategy.** The scarcity of dedicated quad-mesh datasets is addressed by a novel curation pipeline. Starting from diverse 3D sources, we apply automated triangle-to-quad conversion and multi-stage quality filtering to select 1.3 million high-quality models. This dataset is pivotal for training QuadGPT. Further details on our data curation can be found in Appendix A.

## 3.3 TOPOLOGICAL REFINEMENT WITH REINFORCEMENT LEARNING

While pre-training teaches syntactic validity, the cross-entropy loss is a local objective that cannot optimize for global, emergent properties like clean topology. To address this, we introduce a reinforcement learning (RL) stage using Direct Preference Optimization (DPO) (Rafailov et al., 2023) to explicitly align our model with the topological structures preferred in professional 3D workflows.

**Topological Scoring Standard.** Our alignment is guided by a specialized scoring standard designed to evaluate the quality of *generated mesh subsequences*. The reward function quantifies topological integrity by primarily rewarding the formation of long, continuous edge loops ($L_{\text{avg}}$) and penalizing generation fractures ($R_{\text{frac}}$). These metrics are computed automatically, providing a scalable signal for topological quality. The detailed formulation of these metrics is provided in Appendix B.

**Truncated DPO(tDPO)-based Post-Training.** We finetune the pretrained QuadGPT policy by forming preference pairs using our topological rewards and optimizing a DPO-style objective on the collected pairs. Let $\pi_\theta$ denote the trainable policy being fine-tuned, $\pi_{\text{ref}}$ the frozen reference model, and $\beta > 0$ a parameter controlling the deviation from $\pi_{\text{ref}}$. The current policy $\pi_\theta$ produces candidates for input $x$, which are ranked by topological rewards to yield preference pairs $(y_w, y_l)$ with

$$L_{\text{avg}}(y_w) > L_{\text{avg}}(y_l), \tag{6}$$
$$R_{\text{frac}}(y_w) < R_{\text{frac}}(y_l). \tag{7}$$

We assume pairwise preferences follow a Bradley–Terry (BT) likelihood parameterized by an implicit reward $r_\theta(y|x)$. In KL-regularized policy optimization with reference $\pi_{\text{ref}}$, the optimal policy satisfies

$$\pi_\theta(y|x) \; \propto \; \pi_{\text{ref}}(y|x) \, \exp\!\big(r_\theta(y|x)/\beta\big), \tag{8}$$

which implies the implicit reward is (up to an $x$-only constant $c(x)$):

$$r_\theta(y|x) \;=\; \beta[\log \pi_\theta(y|x) - \log \pi_{\text{ref}}(y|x)] + c(x). \tag{9}$$

Substituting equation 9 into the BT model cancels $c(x)$ and yields

$$\mathbb{P}_\theta(y_w \succ y_l \,|\, x) = \sigma\!\left( \beta\Big[ \log \tfrac{\pi_\theta(y_w|x)}{\pi_{\text{ref}}(y_w|x)} - \log \tfrac{\pi_\theta(y_l|x)}{\pi_{\text{ref}}(y_l|x)} \Big] \right). \tag{10}$$

where $\sigma(z) = 1/(1 + e^{-z})$.

To make this process computationally tractable for long sequences, we train on random prefixes of length $m \sim \mathcal{U}\{1, \ldots, L\}$, where $L$ is the total length of the mesh. Let $y_{m:m+\tau}$ denote the sequence from the prefix to the truncation window, where $\tau$ is the window length (e.g., 36,864 tokens), then maximizing the BT likelihood equation 10 gives the tDPO loss

$$\mathcal{L}_{\text{tDPO}}(\theta) = -\mathbb{E}_{\mathcal{D}}\,\mathbb{E}_m \left[ \log \sigma\Big( \beta\Big[ \log \tfrac{\pi_\theta(y_{w,m:m+\tau}|x)}{\pi_{\text{ref}}(y_{w,m:m+\tau}|x)} - \log \tfrac{\pi_\theta(y_{l,m:m+\tau}|x)}{\pi_{\text{ref}}(y_{l,m:m+\tau}|x)} \Big]\Big) \right]. \tag{11}$$

tDPO optimizes each face sequence block $\mathcal{S}(\mathbf{f}^j)$ within the truncation window. This teaches QuadGPT to make locally optimal decisions that lead to globally superior topology. After tDPO optimization, QuadGPT presents a higher quality quad-mesh with more structured edge loops and reduced fractures.

## 4 EXPERIMENTS

### 4.1 EXPERIMENTAL SETTINGS

**Datasets** QuadGPT is pretrained on a curated dataset of 1.3 million quad-dominant models, assembled through an extensive collection, conversion, and filtering pipeline from sources including ShapeNetV2 (Chang et al., 2015), 3D-FUTURE (Fu et al., 2021), Objaverse (Deitke et al., 2023b), Objaverse-XL (Deitke et al., 2023a), and proprietary licensed assets. For preference alignment, we construct a specialized post-training dataset built upon 500 diverse, high-quality dense meshes (including both hard-surface and organic models) generated by Hunyuan3D 2.5 (Lai et al., 2025). After an initial quality filter via rejection sampling, we generate multiple candidates from these source meshes to construct an initial set of approximately 2,000 preference pairs for fine-tuning. This core set of 500 meshes serves as the foundation for DPO. To enhance the model's robustness against noisy, dense inputs, point clouds during pre-training are densely sampled at 40,960 points and augmented with random perturbations.

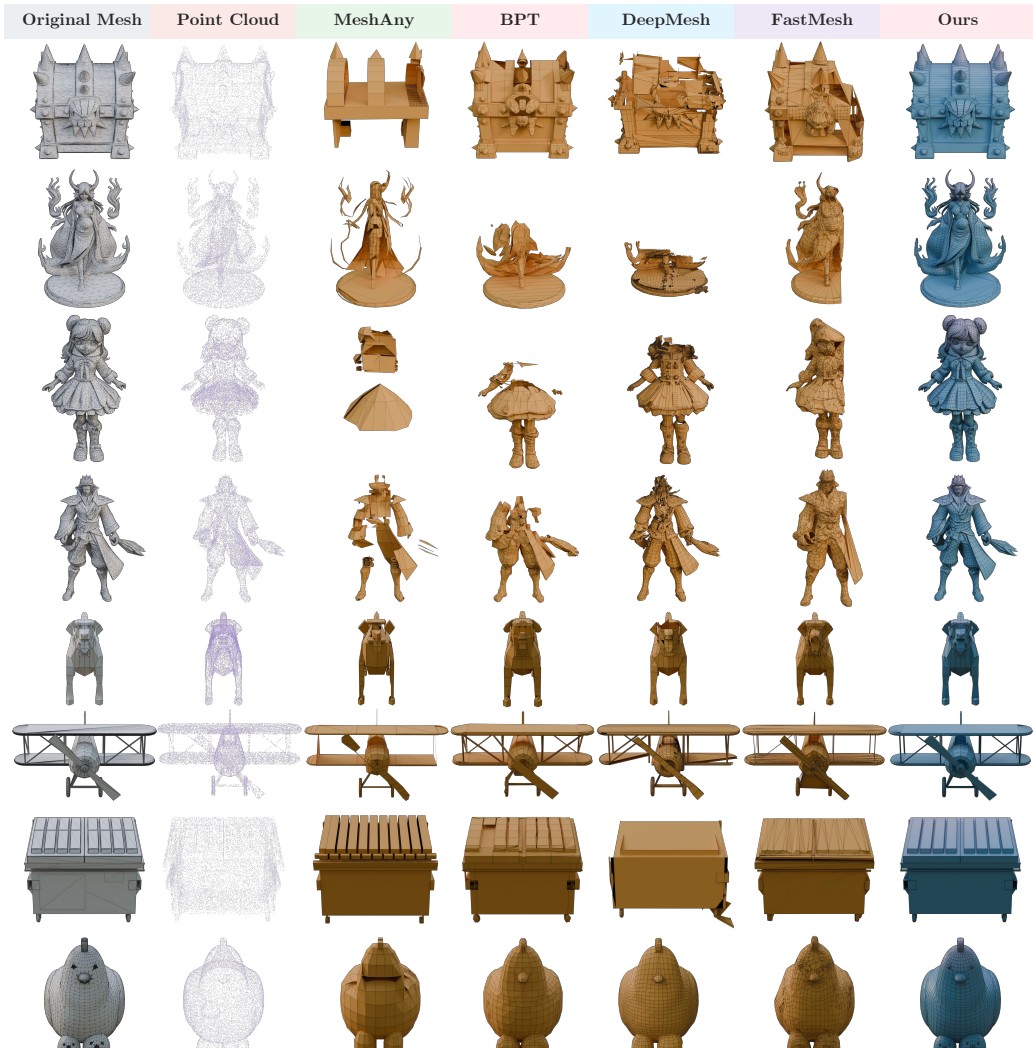

Figure 4: **Qualitative Comparison against Indirect Autoregressive Pipelines.** The top four rows show results on out-of-distribution dense meshes generated by Hunyuan3D (Lai et al., 2025), while the bottom four rows showcase performance on artist-designed meshes. Baseline methods followed by tri-to-quad conversion often produce topological artifacts and lose geometric detail. QuadGPT consistently generates meshes with superior topological coherence and fidelity across both domains.

**Implementation Details**    We pretrain QuadGPT on a cluster of 64 NVIDIA A100 GPUs for 7 days using the AdamW optimizer (Loshchilov & Hutter, 2019) ($\beta_1 = 0.9, \beta_2 = 0.95$) with a learning rate of 1e-4 and a linear warmup schedule. The **Decoder** is a 1.1B parameter model featuring 24 Transformer layers arranged in a three-stage hourglass architecture. This architecture employs linear downsample layers with factors of 4 and 3 to efficiently process long sequences. The subsequent RL fine-tuning stage is performed for 4 hours on the same hardware setup, using a reduced learning rate of 1e-7. Both pre-training and fine-tuning leverage a truncated sequence strategy to manage the long contexts of high-resolution meshes. During inference, the architecture supports a context window of 36,864 tokens. We generate meshes using a combination of top-k and nucleus (top-p) sampling with $k = 10$ and $p = 0.95$, along with a temperature of $T = 0.5$, to balance output diversity and stability. To optimize performance, inference is accelerated through a custom implementation of KV caching and CUDA graphs, specifically tailored for the hourglass architecture. This achieves a generation speed of approximately 230 tokens per second on a single A100 GPU.

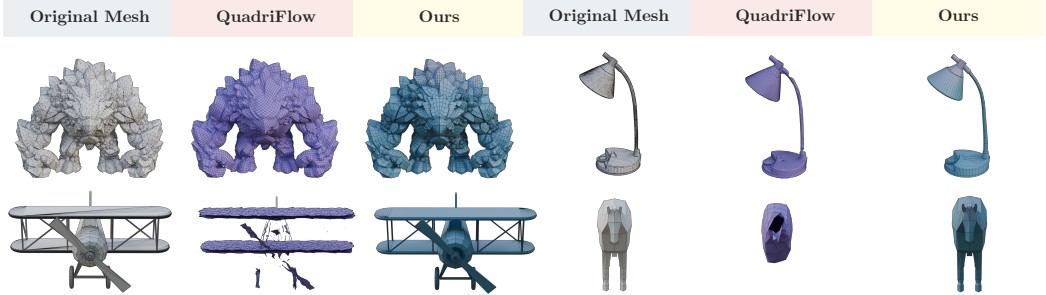

Figure 5: **Qualitative Comparison against a Field-Guided Method.** Field-guided methods like QuadriFlow can be unstable on meshes with complex topology or sharp features.

Table 1: **Quantitative comparison with other baselines in Artist and Dense Meshes.** Our approach achieves superior performance compared to existing baselines. Quadriflow* are computed only on the subset of inputs for which it successfully generated a mesh; its user study score incorporates a score of 0 for all failure cases.

| Data Type | Dense Meshes | | | | Artist Meshes | | | |
|---|---|---|---|---|---|---|---|---|
| Metrics | CD ↓ | HD ↓ | QR ↑ | US ↑ | CD ↓ | HD ↓ | QR ↑ | US ↑ |
| Quadriflow* (Huang et al., 2018) | **0.045** | **0.099** | **100%** | 1.6 | 0.281 | 0.531 | **100%** | 0.3 |
| MeshAnythingv2 (Chen et al., 2025c) | 0.153 | 0.394 | 53% | 1.4 | 0.096 | 0.251 | 60% | 2.1 |
| BPT (Weng et al., 2025) | 0.115 | 0.283 | 43% | 2.7 | 0.051 | 0.125 | 49% | 3.1 |
| DeepMesh (Zhao et al., 2025a) | 0.246 | 0.435 | 64% | 3.3 | 0.236 | 0.417 | 66% | 2.8 |
| FastMesh (Kim et al., 2025) | 0.105 | 0.257 | 3% | 1.1 | 0.052 | 0.141 | 17% | 1.9 |
| **Ours** | 0.057 | 0.147 | 80% | **4.9** | **0.043** | **0.095** | 78% | **4.8** |

**Baselines.** We evaluate QuadGPT against two primary categories of state-of-the-art mesh generation methods. The first category comprises leading autoregressive models that generate triangular meshes, including **MeshAnythingV2** (Chen et al., 2025c), **BPT** (Weng et al., 2025), **DeepMesh** (Zhao et al., 2025a), and **FastMesh** (Kim et al., 2025). Since these methods are fundamentally designed to produce triangular outputs, we apply a robust triangle-to-quadrilateral conversion algorithm like (Muntoni & Cignoni, 2021) as a post-processing step to facilitate a fair comparison. The second category represents specialized quad-meshing techniques, for which we include the well-established field-guided method **QuadriFlow** (Huang et al., 2018). This selection provides a comprehensive benchmark against both the latest in generative modeling and classic, topology-focused approaches.

## 4.2 QUALITATIVE RESULTS

We first present a qualitative comparison of QuadGPT against existing baselines across two distinct domains: challenging, out-of-distribution dense meshes from other AI models, and in-distribution, high-quality artist-designed meshes. Visual inspection is crucial, as it reveals the subtle yet critical differences in topological quality that metrics alone cannot fully capture. As illustrated in Figure 4, the indirect pipeline of converting triangular meshes from autoregressive baselines often struggles. These methods frequently produce meshes with significant topological artifacts, missing geometric details, or overly simplified structures that fail to capture the original shape's nuance. In contrast, QuadGPT consistently generates meshes that are significantly more coherent and artistically plausible, producing the clean edge flow characteristic of professional work. Our model demonstrates strong robustness on challenging AI-generated assets and achieves near-perfect topological reconstruction on artist-designed meshes. Figure 5 compares our method against the field-guided approach of QuadriFlow. The baseline exhibits significant instability on meshes with complex topology or sharp features, often resulting in severe geometric degradation or catastrophic failures. QuadGPT, in contrast, demonstrates exceptional robustness, faithfully reconstructing geometry while maintaining high-quality, structured topology in all examples.

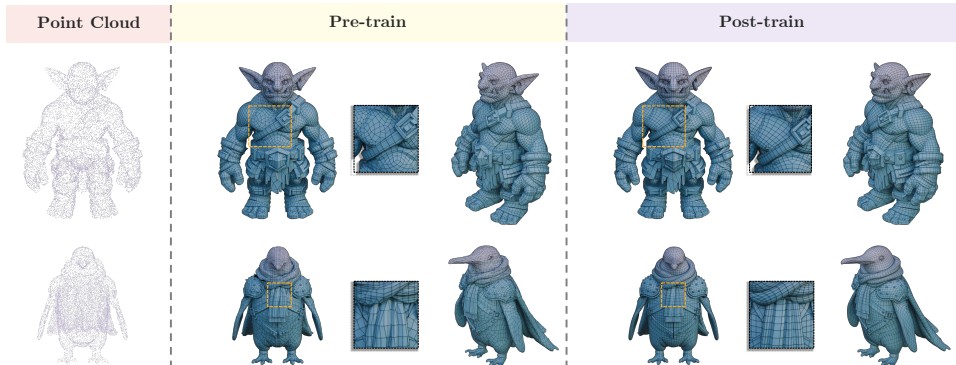

Figure 6: **Effectiveness of tDPO-Pro.** Our comprehensive training strategy significantly enhances both the geometric quality and structural integrity of the generated quad-meshes.

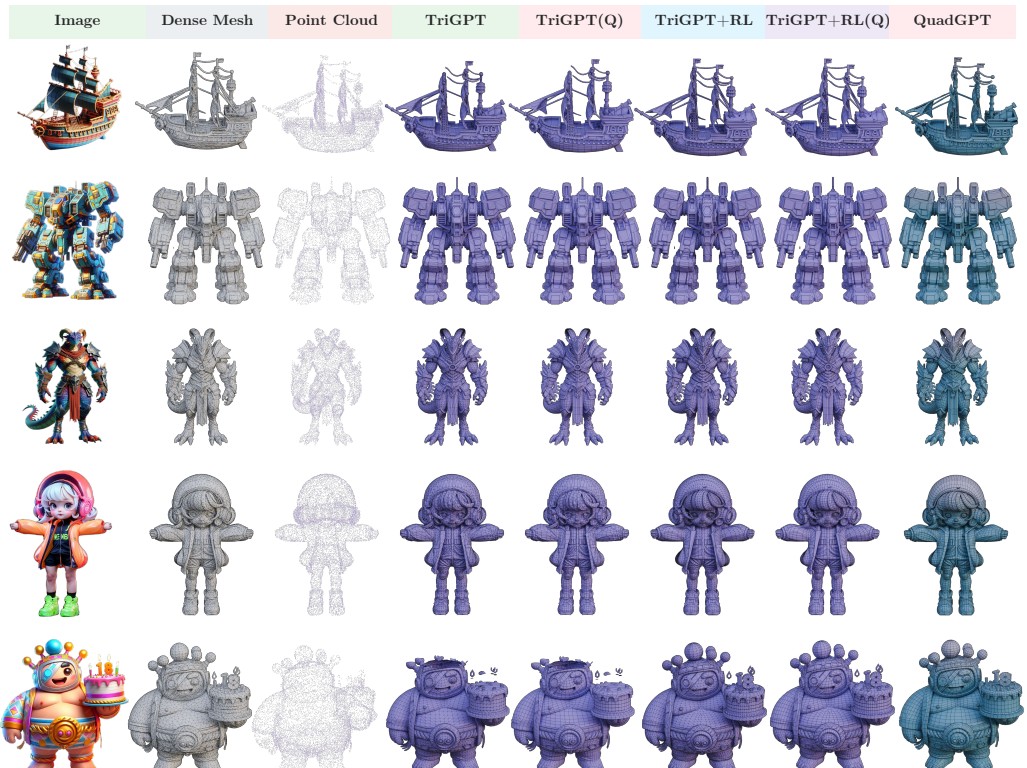

Figure 7: **Native Generation vs. Conversion Pipeline.** (Q) denotes the use of a triangle-to-quad conversion step. Although TriGPT employs the same RL fine-tuning to mitigate fractures, its topological quality is inherently constrained by the post-processing conversion, yielding significantly inferior edge flow compared to our native QuadGPT.

## 4.3 QUANTITATIVE RESULTS

Table 1 summarizes the quantitative comparison of QuadGPT against all baselines across both artist-designed and AI-generated dense meshes. We evaluate geometric fidelity using Chamfer Distance (CD) and Hausdorff Distance (HD), topological quality via Quad Ratio (QR), and perceptual quality through a comprehensive user study (US). The results demonstrate that QuadGPT consistently and significantly outperforms competing approaches in both quality and robustness. The user study, where experts ranked the outputs of all six methods from best (5 points) to worst (0 points), reveals a decisive preference for our method. This confirms that its advantages in producing production-ready assets are not only quantitatively measurable but also perceptually significant.

## 4.4 ABLATION STUDIES

### 4.4.1 EFFECTIVENESS OF tDPO

We analyze the impact of our DPO components by comparing three variants: standard DPO (fine-tuning on full, low-face-count meshes), tDPO (truncated training with a basic fracture penalty), and our full tDPO-Pro model (truncated training with the complete topological reward). Table 2 shows that standard DPO fails to generalize to complex meshes. In contrast, tDPO dramatically improves performance, while tDPO-Pro achieves the best results across all metrics. This improvement, visualized in Figure 6, provides strong empirical support for our comprehensive, topology-aware, truncated DPO framework.

### 4.4.2 EFFECTIVENESS OF CURRICULUM-BASED PRE-TRAINING

We validate our curriculum learning strategy by comparing it against a model trained directly on quad-dominant meshes from scratch ("From Scratch"). As shown in Table 2, the "From Scratch" model struggles to converge, resulting in poor geometric fidelity. This difficulty arises because predicting a quadrilateral face is inherently more complex than predicting a triangle, as it is topologically equivalent to predicting two correlated triangles simultaneously.

Table 2: **Quantitative Evaluation of Training Strategy.** User study scores reflect expert rankings of the five methods, ranging from 0 (worst) to 4 (best).

| Method | CD $\downarrow$ | HD $\downarrow$ | QR $\uparrow$ | US $\uparrow$ |
|---|---|---|---|---|
| From Scratch | 0.081 | 0.203 | 75% | 0.6 |
| Finetune | 0.065 | 0.167 | 72% | 1.3 |
| DPO | 0.073 | 0.188 | 74% | 1.1 |
| tDPO | 0.061 | 0.156 | 78% | 3.3 |
| **tDPO-Pro** | **0.057** | **0.147** | **80%** | **3.7** |

Initializing weights from a converged triangle-generation model ("Finetune") yields significantly superior results. This confirms that our curriculum strategy leveraging the simpler task of triangle generation as a warmup is essential for establishing the stable geometric foundation required to master the more challenging patterns of quadrilateral topology.

### 4.4.3 NATIVE GENERATION VS. CONVERSION PIPELINE

To isolate the benefits of our native approach, we introduce a strong baseline, **TriGPT**, which generates triangles that are then converted to quads. To ensure a fair comparison, TriGPT shares the **identical architecture, 1.3 million mesh training dataset, and tDPO reinforcement learning strategy** as QuadGPT. This setup controls for all confounding variables, testing only the efficacy of the end-to-end native pipeline versus the generation-then-conversion paradigm.

The results in Table 3 and Figure 7 confirm our hypothesis. While the highly optimized triangle baseline (TriGPT+RL) achieves slightly better geometric scores (CD/HD), it cannot match the topological quality of our native approach. QuadGPT demonstrates a substantially higher Quad Ratio (QR) and, crucially, a **user preference score 2.6x higher** than the strongest baseline. This decisive gap in perceptual quality

Table 3: **Native vs. Conversion Pipeline.** QuadGPT is compared against a strong triangle-generation baseline (TriGPT), both with and without RL. US scores are expert rankings (0–2).

| Method | CD $\downarrow$ | HD $\downarrow$ | QR $\uparrow$ | US $\uparrow$ |
|---|---|---|---|---|
| TriGPT(Q) | 0.062 | 0.160 | 70% | 0.2 |
| TriGPT+RL(Q) | **0.051** | **0.138** | 72% | 0.5 |
| **QuadGPT (Ours)** | 0.057 | 0.147 | **80%** | **1.3** |

confirms that while post-hoc conversion struggles to create coherent global structures, our end-to-end native framework excels at learning the artist-preferred topologies, validating its superior practical utility.

## 5 CONCLUSION

We present QuadGPT, the first autoregressive framework that directly generates native quadrilateral and mixed-element meshes. It achieves state-of-the-art results in generative meshing, setting a new standard for both geometric fidelity and topological quality. Our scalable, neural-first approach departs from previous triangle-based methods and conversion pipelines, which rely on heuristic post-processing to approximate quad topology. By leveraging a unified serialization scheme and a novel topology-aware fine-tuning stage (tDPO), QuadGPT directly optimizes for global structure, making it well-suited for the automated creation of production-ready 3D assets.

## 6 ETHICS STATEMENT

This work introduces QuadGPT, an autoregressive framework for direct generation of production-ready, quad-dominant meshes. The model was trained on a blend of public and professionally sourced data, subjected to rigorous quality filtering. Our primary goal is to demonstrate that a scalable, end-to-end framework is effective for industrial-quality mesh generation, and we encourage the community to explore further along this scalable paradigm. To support practical adoption, a public API and online interface will be provided. The authors declare no conflicts of interest.

## 7 REPRODUCIBILITY STATEMENT

This paper presents a scalable autoregressive framework for native quad-mesh generation. Although full model release is not feasible at this stage, the method is comprehensively detailed in terms of data representation, network architecture, and training protocols to facilitate replication and extension. We emphasize that high-quality data curation, including rigorous collection, processing, and filtering of open-source data, is essential for achieving comparable performance. To support validation and downstream use, we will provide a public API and Code.

## ACKNOWLEDGMENTS

This research was supported by fundings from the Hong Kong RGC General Research Fund (152228/23E, 162161/24E, 162116/25E, 162180/25E), National Natural Science Foundation of China (NSFC) Key Program (No.62532005), Collaborative Research Fund (No. C1042-23GF, No. 5097-25G), NSFC/RGC Collaborative Research Scheme (Grant No. 62461160332 & CRS_HKUST602/24), Research Impact Fund (No. R5011-23F), Areas of Excellence Scheme (AoE/E-601/22-R), and the InnoHK (HKGAI).

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

# APPENDIX

## A  DATASET CURATION PIPELINE

The foundation of QuadGPT is a large-scale, high-quality dataset. This section details our three-stage pipeline for its construction: data sourcing and augmentation, multi-stage quality filtering, and final compilation.

### A.1  DATA SOURCING AND AUGMENTATION

The foundation of QuadGPT is a large-scale, high-quality dataset. This section details our three-stage pipeline for its construction: data sourcing and augmentation, multi-stage quality filtering, and final compilation. We aggregate 3D models from diverse sources, including Objaverse-XL Deitke et al. (2023a) and professional modeling repositories. To substantially augment our training data with quad-dominant examples, we developed a **Triangle-to-Quadrilateral Conversion Operator**. Given a triangle mesh $\mathcal{M}_T$, the goal is to find an optimal set of internal edges $E_{\text{int}} \subset \mathcal{M}_T$ to dissolve. We formulate this as an Integer Linear Programming (ILP) problem. For each edge $e \in E_{\text{int}}$, we define a binary decision variable $x_e \in \{0, 1\}$. The optimization objective is:

$$\text{maximize} \quad \sum_{e \in E_{\text{int}}} w_e \cdot x_e \tag{12}$$

subject to the constraint that for any triangle $t \in \mathcal{M}_T$,

$$\sum_{e \in \text{edges}(t)} x_e \leq 1 \tag{13}$$

The weight $w_e$ is a quality score that favors dissolving edges that form well-conditioned quadrilaterals. Constraint 13 ensures each triangle participates in at most one merge operation, preserving manifold integrity. Following this conversion, we perform a crucial **geometric validation** step: any newly formed quadrilateral exhibiting a maximum interior angle greater than 150 degrees is deemed geometrically unstable and is split back into its two original constituent triangles. This ensures our automated pipeline produces only high-quality, well-shaped quadrilaterals.

### A.2  MULTI-STAGE QUALITY FILTERING

The initial dataset, while extensive, contains numerous low-quality models. We employ a rigorous, two-stage filtering pipeline to ensure data fidelity. First, we apply a suite of **rule-based operators** to discard models with specific flaws, including high-aspect-ratio faces and patterns characteristic of poor automated decimation. A critical component is our **Fractured Geometry Detector** (Algorithm 1), which identifies open seams by performing a "test weld" and checking for a significant reduction in edge count without a loss of faces. Second, to filter for aesthetic quality, we trained a **vision-based quality assessment model** on a manually annotated corpus of 100,000 models, allowing us to automate the removal of assets with poor, albeit technically valid, edge flow.

### A.3  FINAL DATASET COMPILATION

Following this pipeline, we selected models with face counts between 500 and 20,000 to form our final training dataset, comprising **1.3 million** high-quality, production-ready 3D models.

---

**Algorithm 1:** Fractured Geometry Detection

---

**Input** : Mesh Object $\mathcal{M}$
**Output:** Boolean 'has_fracture'

$\mathcal{M}'' \leftarrow$ Preprocess($\mathcal{M}$) /* Remove duplicates and loose geometry      */
Components $\leftarrow$ SeparateIntoConnectedComponents($\mathcal{M}''$);
**for** *component in Components* **do**
    **if** *num_vertices(component)* $< \tau_{vtx\_min}$ **then**
        continue;
        /* Ignore small fragments                            */
    **end**
    **if** *not IsManifold(component)* **then**
        $E_{\text{before}}, F_{\text{before}} \leftarrow$ GetEdgeAndFaceCount(component);
        component' $\leftarrow$ MergeVerticesByDistance(component, threshold $= \tau_{\text{weld}}$);
        $E_{\text{after}}, F_{\text{after}} \leftarrow$ GetEdgeAndFaceCount(component');
        **if** $E_{before} - E_{after} > \tau_{edge\_delta}$ ***and*** $F_{before} = F_{after}$ **then**
            **return** true;
            /* Fracture detected                              */
        **end**
    **end**
**end**
**return** false;

---

**Algorithm 2:** Fracture Count Calculation

---

**Input** : Partial Mesh $\mathcal{M}_k = (\mathcal{V}_k, \mathcal{F}_k)$
**Output:** Fracture Count $C_{\text{frac}}$

/* Define generation frontier from the last generated face.   */
$\mathbf{f}_{\text{last}} \leftarrow$ last face in the sequence of $\mathcal{F}_k$;
**if** $\mathbf{f}_{last}$ *is null* **then**
    **return** 0;
**end**
$y_{\text{frontier}} \leftarrow \min_{v \in \mathbf{f}_{\text{last}}}(v.y)$;
/* Count boundary faces at or below the frontier.            */
$\mathcal{F}_{\text{boundary}} \leftarrow$ set of faces in $\mathcal{F}_k$ on the mesh boundary;
$C_{\text{frac}} \leftarrow 0$;
**for** *face f in* $\mathcal{F}_{boundary}$ **do**
    **if** $\forall v \in f, v.y \leq y_{frontier}$ **then**
        $C_{\text{frac}} \leftarrow C_{\text{frac}} + 1$;
    **end**
**end**
**return** $C_{frac}$

---

## B  TOPOLOGICAL QUALITY METRICS FOR TRUNCATED SEQUENCES

This section provides a formal definition of the metrics used to evaluate the topological quality of partially generated quadrilateral meshes. These metrics are specifically designed to operate on truncated sequences, forming the basis of the reward signal for our Direct Preference Optimization (DPO) fine-tuning stage.

### B.1  FRACTURE DETECTION

A primary failure mode in autoregressive generation is fractures. Since our serialization is canonically ordered bottom-to-top, we detect fractures by identifying boundary faces at the current generation frontier. Algorithm 2 formalizes this by defining the frontier based on the lowest Y-coordinate of the last generated face. A non-zero count indicates a failure to generate an adjacent face in the expected upward direction, signaling a topological break.

## B.2 QUANTIFYING EDGE FLOW: QUAD RINGS AND LINES

The hallmark of high-quality quad topology is structured edge flow, which we quantify by identifying two key structures: **Quad Rings** (closed loops of faces) and **Quad Lines** (open strips), as illustrated in Figure 8. Algorithm 3 formalizes an edge-based traversal that "walks" across adjacent quadrilateral faces. If a path terminates, it is classified as a Quad Line; if it returns to its starting edge, it forms a Quad Ring. From the discovered sets of rings ($\mathbb{R}$) and lines ($\mathbb{L}$), we compute the final reward components: the ratio of faces participating in closed rings and the average length of the open lines, both of which are maximized during DPO.

---

**Algorithm 3:** Quad Ring and Line Discovery

---

**Input** : Partial Mesh $\mathcal{M}_k$
**Output:** Set of Quad Rings $\mathbb{R}$ (face lists), Set of Quad Lines $\mathbb{L}$ (face lists)

$\mathbb{R}, \mathbb{L} \leftarrow \emptyset, \emptyset$;
$E_{\text{processed}} \leftarrow \emptyset$;
**for** *edge $e_{start}$ in $\mathcal{M}_k$.edges* **do**
    **if** $e_{start} \notin E_{processed}$ **then**
        path_faces $\leftarrow \emptyset$; path_edges $\leftarrow \emptyset$;
        current_edge $\leftarrow e_{\text{start}}$;
        **while** *current_edge $\neq$ null **and** current_edge $\notin$ path_edges* **do**
            add current_edge to path_edges;
            adj_quad $\leftarrow$ a quadrilateral face adjacent to current_edge;
            **if** *adj_quad exists* **then**
                add adj_quad to path_faces;
                current_edge $\leftarrow$ edge opposite to current_edge in adj_quad;
            **end**
            **else**
                current_edge $\leftarrow$ null;
            **end**
        **end**
        $E_{\text{processed}}$.add(path_edges);
        `/* Classify the discovered path based on closure.     */`
        **if** *current_edge is null **or** current_edge $\neq e_{start}$* **then**
            add path_faces to $\mathbb{L}$;
        **end**
        **else**
            add path_faces to $\mathbb{R}$;
        **end**
    **end**
**end**
**return** $\mathbb{R}, \mathbb{L}$

---

## C MORE RESULTS

To further demonstrate the superiority of our direct generation approach, we provide extensive qualitative comparisons in Figure 9 and Figure 10. These results highlight the robustness and high fidelity of QuadGPT across a wide variety of complex geometries. Additionally, to illustrate the structural integrity of our generated models, we present multi-view renderings of several samples in Figure 11. Finally, we compare our method with closed-source, commercial quad mesh generation techniques, including Tripo Team (2025) and Quad Remesher Exoside (2019) in Figure 12.

## D ADDITIONAL EXPERIMENTAL ANALYSIS

In this section, we provide further analysis to complement the experiments in the main paper.

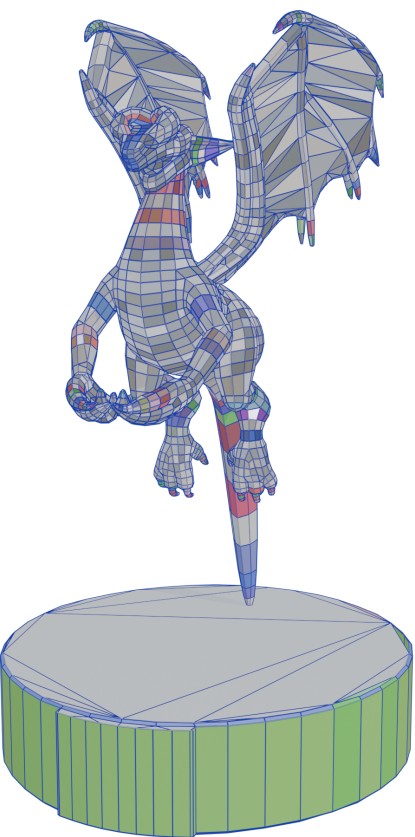

Figure 8: An illustration of a Quad Ring which is closed loop of quadrilateral faces. Our tDPO reward is designed to encourage the formation of such desirable, structured patterns.

### D.1    Impact of Curriculum Learning and Topology-Aware Rewards

Figure 13(a) compares the next token prediction loss curves between from-scratch quadrilateral mesh training and fine-tuning from pre-trained triangle mesh weights. Fine-tuning achieves faster convergence and lower loss, confirming that inheriting triangle mesh knowledge provides better initialization for quadrilateral generation. Figure 13(b) presents the reward curves during DPO training, where our tDPO-Pro approach with truncated edge loop optimization consistently achieves the highest reward, validating the effectiveness of our complete topological reward design. The progressive improvement from DPO to tDPO and finally tDPO-Pro demonstrates the cumulative benefits of our methodological contributions.

### D.2    The Inherent Limitations of Triangle-to-Quad Conversion

To illustrate the limitations of any post-hoc conversion pipeline, we conduct a best-case scenario experiment. We begin with a high-quality artist-created quad mesh and triangulate it. This process represents an irreversible information loss: the artist's original topological intent becomes ambiguous, creating a complex combinatorial problem that heuristic algorithms struggle to solve.

As shown in Figure 14, we compare our own **ILP-based operator** against the default methods in MeshLab Tarini et al. (200) and Blender Blender Foundation. While our optimized operator performs best, none of the methods can perfectly recover the original edge flow, leaving behind topological artifacts.

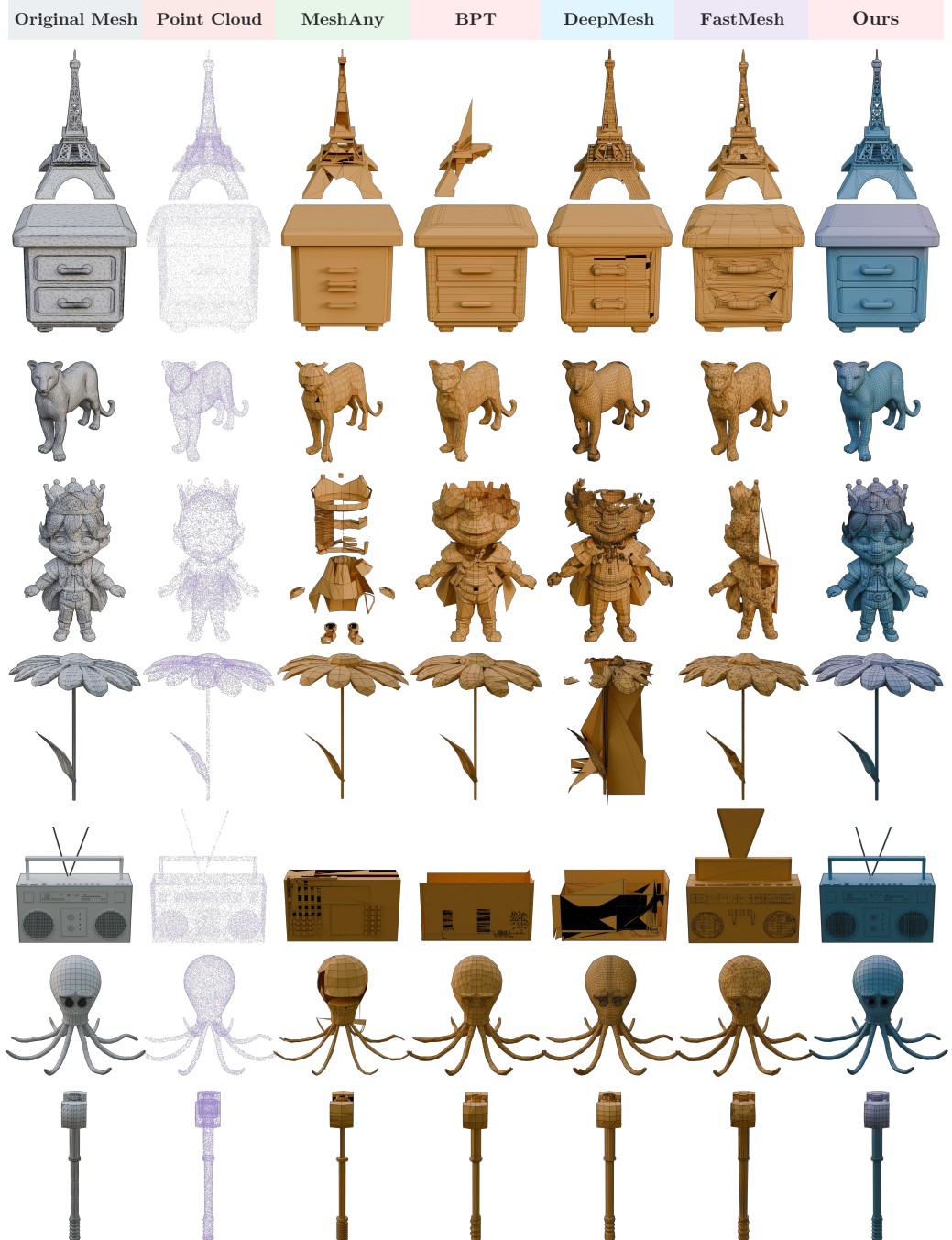

Figure 9: **Extended Qualitative Comparison against triangle generation.**

This challenge is significantly amplified when the input is an AI-generated triangle mesh, which often has an edge flow unsuitable for conversion, as seen in our TriGPT experiments (Figure 7). This compounding effect of a suboptimal input and an imperfect algorithm reinforces the need for our native generation approach.

Furthermore, algorithms from adjacent fields like Blossom-Quad (**?**), designed for 2D finite element analysis, are too restrictive for complex 3D surfaces. Its perfect-matching requirement caused it to fail on all but our simplest test cases, underscoring the need for a domain-specific, learning-based solution like QuadGPT.

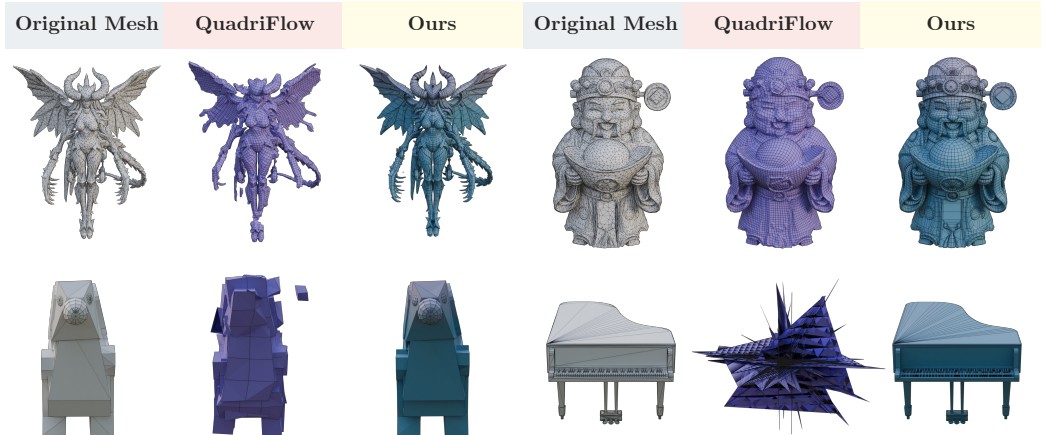

Figure 10: A qualitative comparison against the field-guided method QuadriFlow. This example highlights the superior robustness and topological quality of our direct generative approach.

## D.3 ANALYSIS OF COMPRESSED VS. DIRECT COORDINATE REPRESENTATION

A critical design choice in autoregressive mesh generation is the tokenization strategy. Methods like DeepMesh (Zhao et al., 2025a) and BPT (Weng et al., 2025) employ compressed representations. Our experiments with these methods on triangle meshes confirmed a similar finding: compressed tokenizers lead to faster initial convergence due to shorter sequence lengths. However, we argue that faster convergence does not necessarily lead to better final performance. As shown in Table 4, our extensive experiments at the 1B-parameter scale show that a simple, **direct coordinate representation ultimately achieves superior results** in both geometric fidelity and topological quality, a finding strongly corroborated by our user study. We hypothesize that overly complex, "over-designed" tokenizers can introduce an unintended inductive bias, constraining the model's expressive capacity and limiting its performance ceiling, especially when trained on large, diverse datasets. This observation aligns with a broader trend in deep learning, demonstrated by models like DepthAnything V3 (Lin et al., 2025), where simple, highly scalable architectures often outperform more complex designs when sufficient data and compute are available. For QuadGPT, we therefore prioritized the higher performance ceiling of the direct representation, confident that techniques like truncated training effectively manage the longer sequence lengths.

## D.4 ABLATION ON TRAINING DATA

To disentangle the impact of our methodology from our curated dataset, we conduct a data ablation study. We train a variant, **QuadGPT-OS**, using only the publicly available portions of our dataset (e.g., ShapeNet Chang et al. (2015), Objaverse-XL Deitke et al. (2023a)), subjected to the same rigorous filtering pipeline described in Section A. This experiment serves to answer a critical question: can a robust, scalable methodology achieve state-of-the-art results even when limited to open-source data?

Table 4: **Quantitative Comparison: Tokenization on Triangle Meshes.** Direct coordinate representation (TriGPT) outperforms a compressed BPT-style tokenizer (TriGPT+BPT). User Study scores reflect expert rankings from 0 (worst) to 1 (best). (Q) denotes outputs were converted to quads for evaluation.

| Method | CD $\downarrow$ | HD $\downarrow$ | QR $\uparrow$ | US $\uparrow$ |
|---|---|---|---|---|
| TriGPT+BPT (Compressed) | 0.078 | 0.198 | 68% | 0.3 |
| TriGPT (Direct Coord.) | **0.062** | **0.160** | **70%** | **0.7** |

The results, shown in Figure 15, are twofold and unequivocal. First, **QuadGPT-OS still significantly outperforms all prior autoregressive baselines**. This confirms that our core contributions—the scalable architecture, curriculum learning strategy, and tDPO refinement—are the primary drivers of our model's high performance. It proves that our method is not merely a product of better data, but a fundamentally more solid approach to the problem.

Second, our full model, trained on the complete 1.3 million mesh dataset, demonstrates a further substantial improvement in quality over QuadGPT-OS. This highlights a key insight: while a superior methodology can establish a new state of the art on public data, the full potential of a truly scalable approach is unlocked through careful, large-scale data curation. Our work demonstrates the

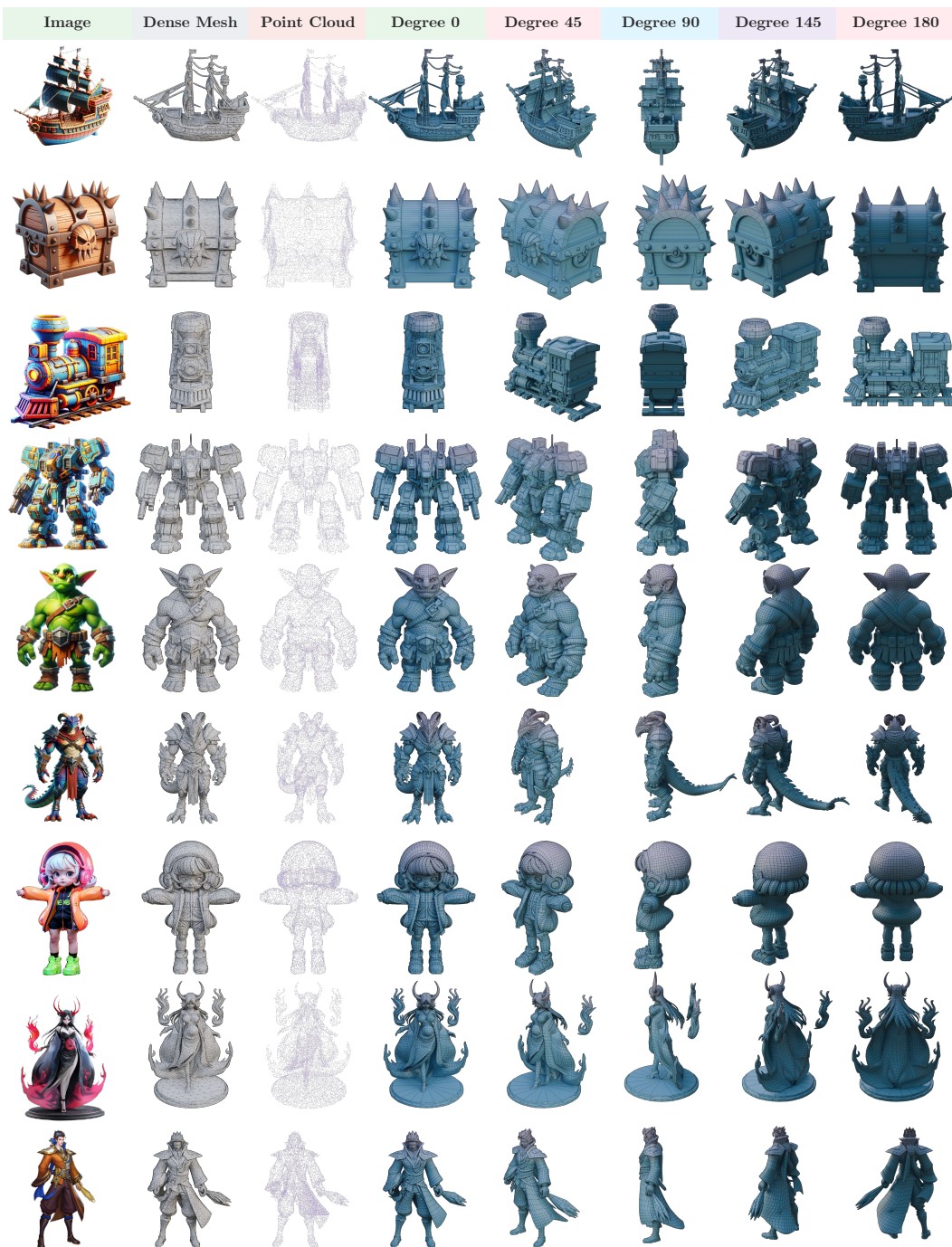

| Image | Dense Mesh | Point Cloud | Degree 0 | Degree 45 | Degree 90 | Degree 145 | Degree 180 |
|---|---|---|---|---|---|---|---|

Figure 11: **Multi-View Renderings of Generated Meshes.**

synergistic impact of advancing both the model and the data to push the frontier of production-ready 3D generation.

## E    DOWNSTREAM APPLICATIONS

This section demonstrates the advantages of high-quality topology in downstream applications, such as UV mapping and animation. Thanks to our superior quad mesh structure, the models facilitate

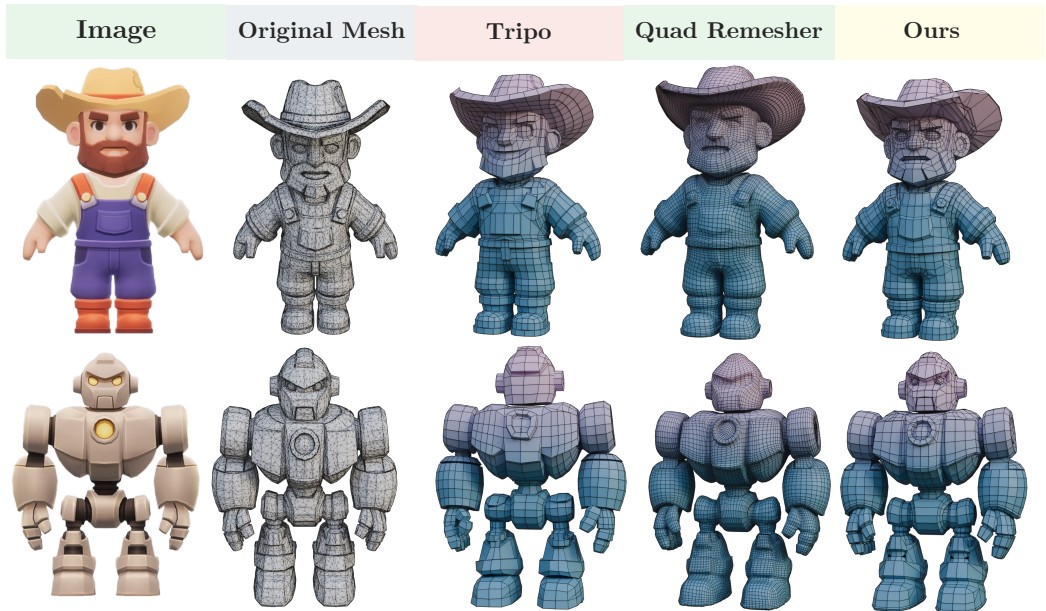

Figure 12: **Comparison with closed-source commercial quad mesh generation methods.** Our approach preserves more geometric details, especially edge loops, which are crucial for UV segmentation and animation. Compared to Blender's remeshing plugin, our method produces compact faces.

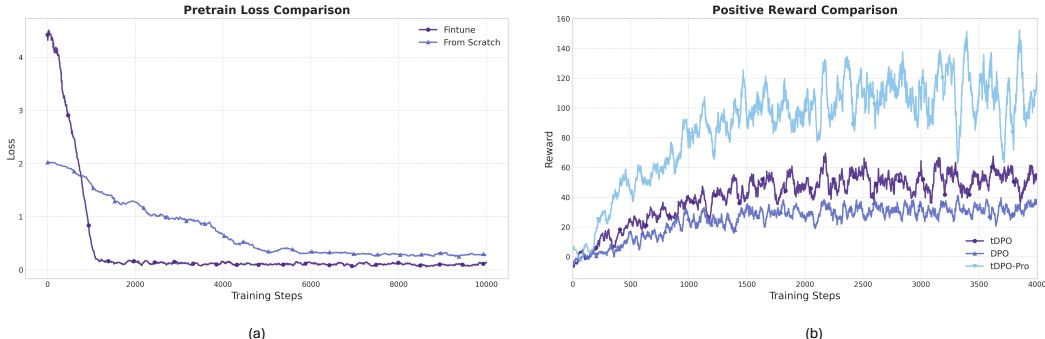

Figure 13: **Training Dynamics Analysis.** (a) Next token prediction loss comparison between from-scratch quadrilateral mesh training and fine-tuning from pre-trained triangle mesh weights. Fine-tuning demonstrates faster convergence and lower loss. (b) DPO reward curves for different variants, showing that our tDPO-Pro approach achieves the highest reward with truncated edge loop optimization.

high-quality UV unwrapping Li et al. (2025a). Furthermore, the abundance of edge loops enables more natural mesh deformation.

## F    LIMITATION AND FUTURE WORK

Despite the promising results of QuadGPT, several limitations present important avenues for future research.

**Domain Gap and Part-based Generation.**    Our framework operates within the prevailing two-stage paradigm, where topology generation follows geometry generation. This creates a critical domain gap: the model is trained on point clouds from clean, artist-crafted assets but is often deployed on noisy point clouds derived from AI-generated implicit fields. This mismatch can lead to fractures and other failures on complex, out-of-distribution shapes, as shown in Figure 17. A

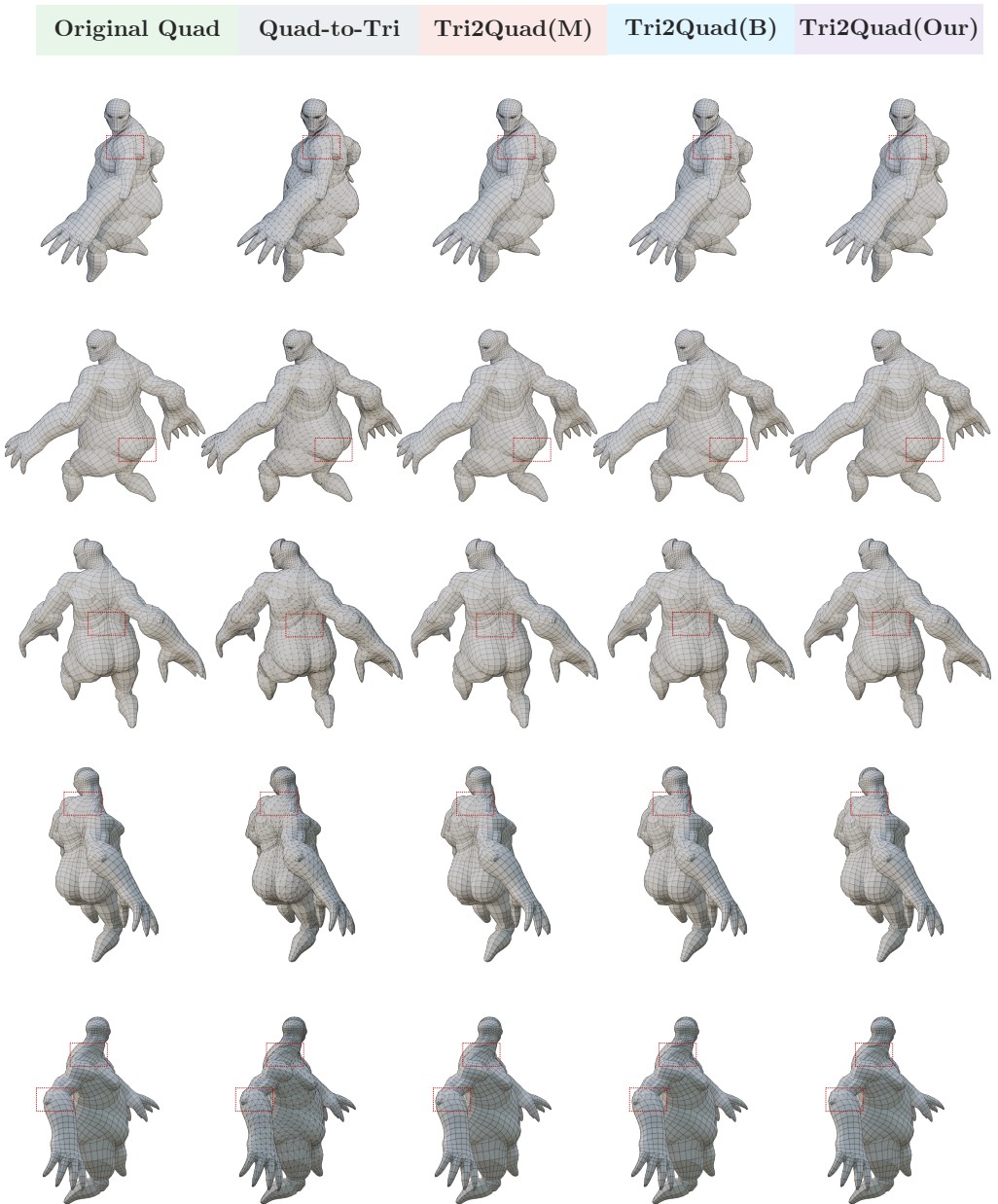

Figure 14: **The Irreversibility of Triangulation and Limitations of Conversion Algorithms.** (a) A high-quality artist-created quad mesh with clean edge flow. (b) The same mesh after triangulation—the ground truth topological information is now ambiguous. (c-e) The results of applying different tri-to-quad conversion algorithms to (b). While our ILP-based operator (e) produces a better result than the default methods in MeshLab (c) and Blender (d), none can perfectly recover the original topology, resulting in broken edge loops and artifacts. This demonstrates the fundamental limitations of post-hoc conversion.

promising direction is to develop an end-to-end model that co-generates geometry and topology, bypassing the intermediate dense mesh entirely. Furthermore, since professional assets are inherently part-based, integrating part-aware generation, potentially building on recent advances in part generation (Yan et al., 2025), could significantly improve robustness and fidelity.

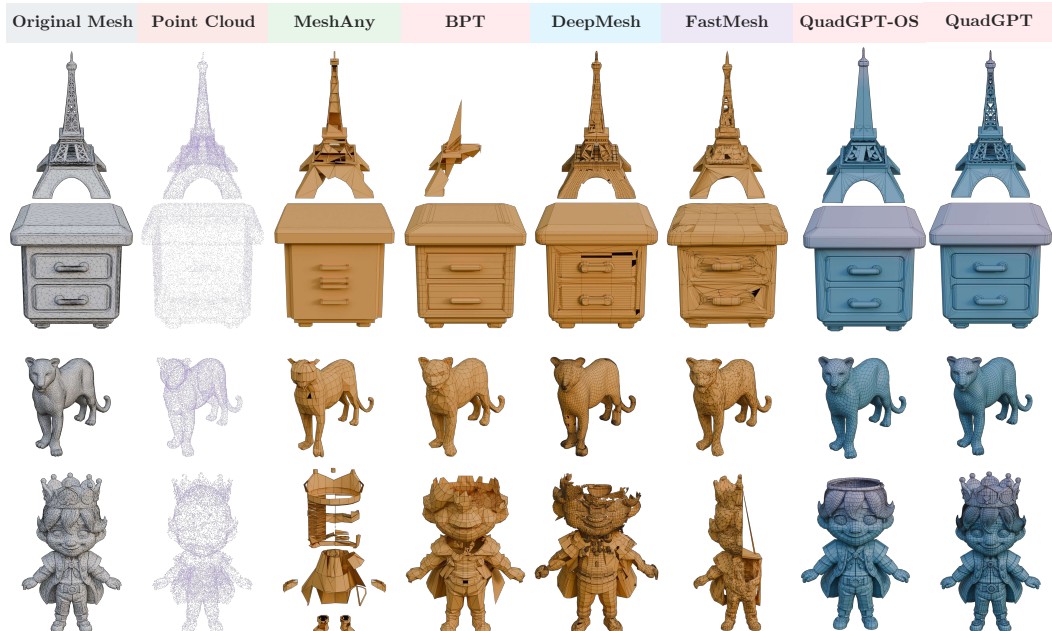

Figure 15: **Qualitative results of our data ablation study.** "QuadGPT-OS" is trained exclusively on filtered public datasets, while "QuadGPT" uses our complete curated dataset. "QuadGPT-OS" already demonstrates a significant improvement in topological coherence and geometric detail over prior baselines. The full model shows further refinement, particularly in achieving cleaner, more professional edge flow, validating the scalability of our approach with high-quality data.

**Lack of Controllable Polygon Number.** QuadGPT currently cannot explicitly control the final polygon count; attempts to add a simple number condition were ineffective. This limits its utility in production, where generating assets with specific polygon budgets for LODs is essential. Future work must develop a robust control mechanism.

**Advancing the Reinforcement Learning Framework.** Our current implementation uses off-policy DPO for its simplicity and stability. However, our programmatic reward, while effective at enforcing structural rules like edge loops, cannot capture the full nuance of artistic preference. Future work could explore two key directions. First, developing a reward model trained on human preference data from professional artists could provide a richer, more aligned optimization signal. Second, investigating more advanced, potentially online RL algorithms could further elevate the quality and complexity of the generated topology.

## G  LLM USAGE STATEMENT

The authors used Gemini 2.5 Pro exclusively for grammar checking and language polishing of the manuscript text. All technical content, experimental design, data analysis, and scientific conclusions are the original work of the authors. The LLM was not involved in generating scientific ideas, conducting experiments, or interpreting results.

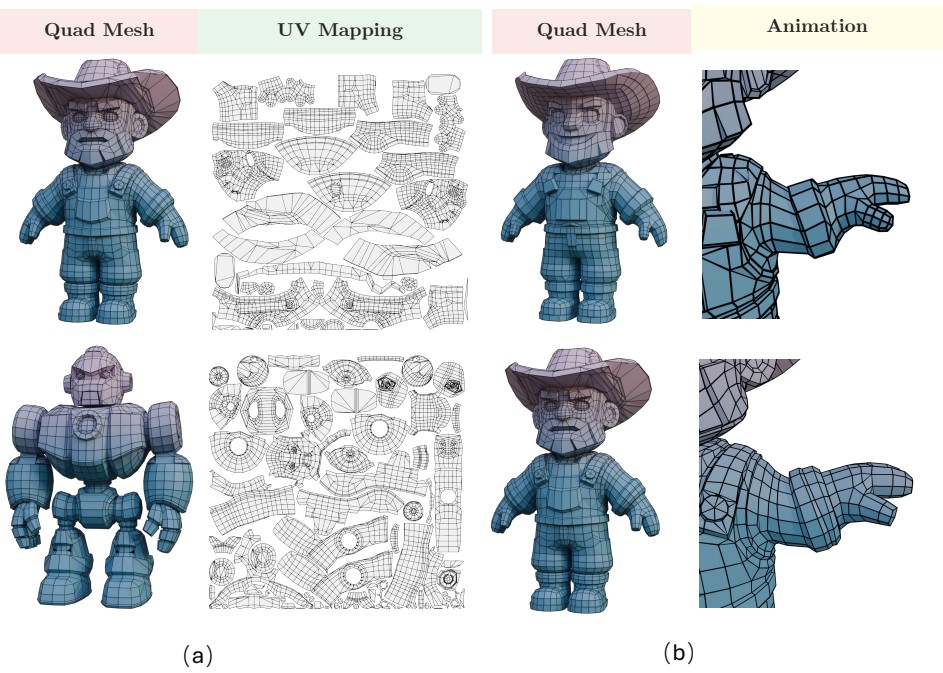

| Quad Mesh | UV Mapping | Quad Mesh | Animation |
|---|---|---|---|

(a)              (b)

Figure 16: **UV mapping and animation deformation comparison.** (a) UV unwrapping result on a quad mesh generated by our method. (b) Animation deformation comparison: the top row shows results on a mesh from Tripo, while the bottom row presents ours. Our method captures geometric details more effectively and produces more edge loops, leading to noticeably more realistic deformation.

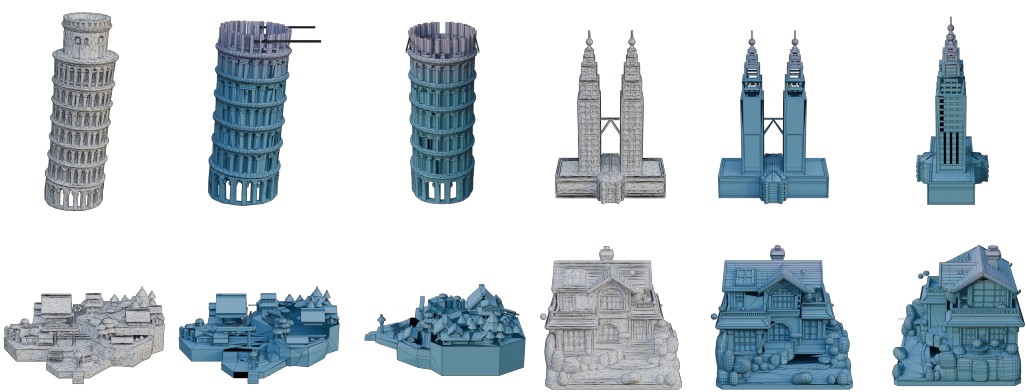

Figure 17: **Failure Cases and Limitations.** Our model can sometimes struggle to generalize to out-of-distribution point clouds sampled from AI-generated assets, particularly for architectural models with sharp features, which can result in fractures or incomplete geometry.

