# OpenReview forum: "QuadGPT: Native Quadrilateral Mesh Generation with Autoregressive Models"
_ICLR.cc/2026/Conference — ICLR 2026 Poster_

### Official Review · Reviewer_N2bc · 2025-10-24

**Soundness:** 3
**Presentation:** 3
**Contribution:** 3
**Rating:** 6
**Confidence:** 4

**Summary:**

The paper introduces QuadGPT, an end-to-end autoregressive framework for generating native mixed quadrilateral and triangular meshes directly from a point cloud input. QuadGPT proposes a unified serialization scheme that handles both triangle and quad faces using a padding-based tokenization, along with an Hourglass Transformer architecture for efficient processing of long sequences. Furthermore, the model is refined using a reinforcement learning approach, which uses a topological reward function to encourage the formation of clean, production-ready edge loops. The authors demonstrate that this approach significantly surpasses prior state-of-the-art methods in both geometric fidelity and topological coherence on a large, curated dataset.

**Strengths:**

- The paper is clearly written and easy to follow.
- The experimental results are impressive, demonstrating strong generalization capability on a wide range of meshes.

**Weaknesses:**

- The method itself is a combination of previous efforts (e.g., hourglass transformer and quad-dominance control form Meshtron, direct mesh tokenization from MeshXL, point cloud encoder from MeshAnything), with the biggest difference as the introduction of a mixed quad-triangle setting, which is a rather simple extension.
- There is no ablation on the dataset. It's hard to tell if the performance boost is mainly from better data quality, and the comparison with previous works trained on public datasets is unfair.
- Missing references and discussions on the triangle-to-quad conversion algorithm, for example, Blossom-Quad [1] and Blender's built in algorithm.

[1] Blossom-Quad: A non-uniform quadrilateral mesh generator using a minimum-cost perfect-matching algorithm; Remacle, J‐F., et al.

**Questions:**

- The padding-based serialization seems pretty plain and inefficient for meshes with many triangles. Have the authors considered about using token compression techniques like BPT?
- I wonder if the authors have tried experiments on openly available dataset? It's unfair to compare with other models trained on different datasets (of potentially worse quality). It's understandable to use proprietary datasets for the best performance, but the author should at least do some ablation study on the dataset quality, which can provide some insights for future research (e.g., the TripoSG paper).
- How effective is the quad-dominance parameter for controlling the ratio of quad vs triangular faces? It also sounds pretty empirical to gradually anneal the data distribution with r from 0 to 1.
- What's the insight for native quad mesh generation? Especially, I wonder its difference from first generating pure triangular meshes with the same model (e.g. use r=0), and then apply the proposed triangle-to-quad algorithm.

---

> ### Author Response · Authors · 2025-11-21
>
> We thank you for your constructive feedback. We have revised the paper to address your concerns regarding dataset ablation, baselines, and technical clarifications.
>
> #### **1. On Novelty: A Holistic Solution, Not a Simple Extension**
> We argue that our contribution goes beyond combining modules, addressing unique challenges in **training stability** and **topological consistency** that triangle-based methods do not encounter.
> *   **Systematic Design:** Our framework is synergistic: **Padding** ensures precise coordinate control, **Curriculum Learning** stabilizes the difficult quad distribution learning, and **tDPO** refines topological quality.
> *   **Course Correction:** Crucially, we offer a "course correction" against the trend of "token compression" (e.g., MeshAnything), demonstrating that **direct coordinate representation** is scalable and avoids the inductive biases of compression that limit topological precision (see **Appendix D.3**).
>
> #### **2. Dataset Ablation & Fair Comparison**
> To disentangle data quality from method effectiveness, we added two rigorous experiments:
> *   **QuadGPT-OS (Open Source Data Only):** We trained a variant using *only* filtered public datasets. As shown in **Figure 15** (Appendix D.4), **QuadGPT-OS still significantly outperforms prior baselines**, confirming that our architecture drives performance, not just proprietary data.
> *   **TriGPT + Converter (Same Data Control):** To isolate the benefits of *native generation*, we trained a triangle-only model (TriGPT) on our full dataset plus conversion. **QuadGPT** still achieves superior edge flow (1.3 vs 0.5, **Table 3**), proving that native generation yields better topology than post-processing.
>
> #### **3. Padding vs. Token Compression**
> We chose fixed-length padding over compression (e.g., BPT) for two reasons:
> 1.  **Topological Precision:** Compression introduces undesirable inductive biases. Our experiments (Appendix D.3) show that it degrades fine-grained topological capture (e.g., edge loops), whereas padding preserves sequence fidelity.
> 2.  **Architectural Synergy & Weight Inheritance:** The **Hourglass Transformer** handles long sequences efficiently. Crucially, padding allows seamless weight inheritance from stable triangle models (Section 4.4.2), enabling robust geometric grounding before learning complex topology.
>
> #### **4. The Role of Parameter `r` and Curriculum Learning**
> We clarify that `r` is a **binary indicator** (0 or 1) for our **curriculum learning strategy**, not a continuous controller.
> *   **Curriculum Learning:** We anneal a sampling threshold from 0 to 1. When $p < \text{threshold}$, we use native quads (`r=1`); otherwise, we use triangles (`r=0`), allowing the model to master geometry before topology.
> *   **Inference Behavior:** During inference, we set `r=1` to generate quad-dominant meshes. We avoid rigid ratios, letting the model adaptively determine the optimal distribution (~80%) to meet professional standards.
>
> #### **5. Insight: Native Generation vs. Post-Processing**
> We have added discussions regarding **Blossom-Quad** and **Blender**. Our insight is that **topology generation is a global planning problem**, not a local conversion task.
> *   **Irreversible Information Loss:** As detailed in **Appendix D.2**, triangulation is often irreversible. Our "round-trip" experiment shows that no converter (Blender, ILP) could perfectly recover the original edge flows due to combinatorial ambiguity.
> *   **Global vs. Local:** Conversion algorithms are local and propagate input irregularities. In contrast, **QuadGPT** learns the *distribution* of professional topologies, jointly planning geometry and topology to align globally with curvature (Section 4.4.3).
> *   **Limitations of Existing Tools:** Algorithms like **Blossom-Quad** (for 2D FEA) are often too restrictive for 3D surfaces, reinforcing the need for our robust, learning-based solution.

---

> ### Author Response · Authors · 2025-11-27
> **A Kind Reminder on the Author Rebuttal**
>
> Dear Reviewer,
>
> Thank you again for your time and insightful feedback. We have posted our author rebuttal, which includes detailed responses and several new experiments conducted to directly address the points raised in the reviews. We would be grateful if you could take a look at our response when you have a moment. We are happy to provide any further clarifications.

---

### Official Review · Reviewer_SA17 · 2025-10-25

**Soundness:** 3
**Presentation:** 4
**Contribution:** 2
**Rating:** 4
**Confidence:** 4

**Summary:**

This work introduces QuadGPT, an autoregressive framework for direct generation of quadrilateral meshes. The input condition is a point cloud with normals. The authors propose a unified representation through padding, supporting mixed-element topologies (triangles + quads). The model is built upon an hourglass architecture. The pretraining loss is a standard cross-entropy loss. The model is further finetuned by a reinforcement learning approach with truncated direct preference optimization, rewarding coherent edge loops. Experiments show that QuadGPT generates higher quality quad meshes when compared to prior methods, both quantitatively and qualitatively.

**Strengths:**

- This work introduces an end-to-end learning-based framework for direct quad mesh generation from point clouds. This is challenging as meshes have complex structures with significantly large numbers of face and vertex elements, and forming coherent edge loops as in professional-crafted quad meshes is difficult.
- To promote clean topologies in the generated quad meshes, the authors introduce a reinforcement learning stage, optimizing a direct preference optimization objective rewarding long, coherent edge loops. To handle long sequences, the authors use a truncated, local window-based approach.

**Weaknesses:**

- The novelty of the proposed method is somewhat limited: the straightforward padding in the sequence representation to support triangles + quads, the hierarchical hourglass architecture from MeshTron [Hao et al. 2024], the direct preference optimization already proposed for mesh generation in DeepMesh [Zhao et al. 2025]. Overall, the proposed method seems to be a simple extension of those existing works to quad mesh generation.
- The experimental evaluation is less comprehensive. Comparisons to the triangle mesh generation baselines (e.g., MeshAnything, DeepMesh) may not be fair, due to the difference in training data and model capacity for long sequences (Fig 4).
- The proposed model has less controllability over the ratio of triangles and quads in the output mesh, though a conditioning mechanism with a quad-dominance parameter is introduced.

**Questions:**

- As mentioned in the weaknesses, the comparisons to triangle mesh generation baselines need to be strengthened. For example, in the training strategy, the authors already pretrained a model exclusively on triangle meshes (L247-248). Combining this model with triangle-to-quadrilateral conversion could be a strong baseline, reducing the difference of training data and model capacity used in other triangle mesh generation baselines.
- The authors introduced a learnable embedding for a quad-dominance parameter to control the target ratio of face types. However, L990 seems to indicate that this is not effective in practice, and there is no corresponding quantitative analysis.
- In L111, the authors claim that QuadGPT bridges the gap between text/image inputs and production-ready 3D artist meshes. However, the majority of the results in the main text are generated from point clouds.
- There is no promise of code and data release. Though the authors mention that a public API will be provided, access to the full model weights and training data is important for reproducibility and follow-up research.

---

> ### Author Response · Authors · 2025-11-21
>
> We thank you for your thorough and insightful review. Your suggestions, particularly regarding the experimental evaluation, are invaluable and have pushed us to provide a more rigorous validation of our core claims.
>
> #### **1. On the Novelty of the Proposed Method**
>
> While we leverage established architectures like the Hourglass Transformer, our contribution represents the **first successful and scalable framework for native quadrilateral mesh generation**, solving critical topological challenges that hinder practical 3D modeling.
>
> *   **A Holistic Solution to a New Problem:** The transition from triangle to quad generation introduces fundamental challenges in topological consistency and training stability. Our contribution is the **synergistic integration of novel strategies** to overcome these hurdles:
>     1.  **Deliberate Design Philosophy:** We intentionally chose a simple, padding-based representation for scalability. As detailed in **Appendix D.3**, this approach allows for seamless weight inheritance and achieves a higher performance ceiling than compressed schemes.
>     2.  **Curriculum Learning:** Since naive training on quads fails, our novel curriculum strategy leveraging pre-trained triangle knowledge is a key innovation essential for convergence.
>     3.  **Topology-Aware Optimization:** Our tDPO strategy is specifically tailored to reward edge loops and singularities characteristic of high-quality quad meshes, capturing nuances that standard RL approaches miss.
>
> #### **2. On Ensuring Fairer Experimental Comparisons**
>
> To definitively isolate our method's contribution from dataset quality, we have introduced two rigorous experiments:
>
> *   **Ablation Study: QuadGPT-OS (Open Source Data Only)**
>     We trained **QuadGPT-OS** using *only* publicly available data (e.g., ShapeNet, Objaverse-XL).
>     **Result:** As shown in **Figure 15** (Appendix D.4), **QuadGPT-OS still significantly outperforms all prior autoregressive baselines**, confirming that our core contributions (architecture, curriculum, tDPO) drive performance independent of proprietary data.
>
> *   **New Baseline: "TriGPT + Converter"**
>     We trained a model with the **exact same architecture and dataset**, but tasked only with generating triangles (TriGPT), followed by high-quality conversion. This setup perfectly isolates the performance gain of our **native approach**.
>     **Result:** As shown in **Table 3** and **Figure 7**, native generation is significantly superior in topological edge flow and user preference (1.3 vs 0.5), proving our advantage comes from the method.
>
> *   **Comparison with Larger Datasets:** Furthermore, even strong baselines like **BPT**, which utilize a **larger dataset (1.5M vs. our 1.3M)**, fail to achieve comparable topological quality, reinforcing that our performance is driven by methodology rather than data scale.
>
> #### **3. On the Effectiveness of the Quad-Dominance Parameter**
>
> We clarify that `r` functions as a **binary indicator** (0 or 1) to switch between mesh representations, not a continuous controller.
>
> *   **Curriculum Learning:** The primary role of `r` is to enable our **curriculum learning strategy**, annealing a sampling threshold from 0 to 1. When $p < \text{threshold}$, we use the native quad representation (`r=1`); otherwise, we use the triangulated representation (`r=0`), allowing the model to master geometry before topology.
> *   **Inference Behavior:** During inference, we set `r=1` to generate quad-dominant meshes. We avoid enforcing a rigid ratio, as our model adaptively determines the optimal distribution (typically ~80%) to meet professional standards.
>
> #### **4. On the Claim Regarding Text/Image Inputs**
>
> You are correct that our model takes point clouds as input. The "gap" refers to the inability of existing generators to robustly handle noisy point clouds from Text/Image-to-3D pipelines. QuadGPT demonstrates exceptional robustness here, serving as a reliable "geometry engine" to convert these representations into production-ready meshes.
>
> #### **5. On Code and Model Release**
>
> We are committed to making our work accessible. We will release the **codebase**, **pre-trained weights**, and a **substantial portion of our dataset**. Additionally, we will provide a public API and interactive demo to facilitate immediate reproducibility without requiring heavy compute.

---

> ### Author Response · Authors · 2025-11-27
> **A Kind Reminder on the Author Rebuttal**
>
> Dear Reviewer,
>
> Thank you again for your time and insightful feedback. We have posted our author rebuttal, which includes detailed responses and several new experiments conducted to directly address the points raised in the reviews. We would be grateful if you could take a look at our response when you have a moment. We are happy to provide any further clarifications.

---

### Official Review · Reviewer_cNgg · 2025-10-30

**Soundness:** 3
**Presentation:** 3
**Contribution:** 2
**Rating:** 4
**Confidence:** 3

**Summary:**

1 The paper extends prior triangle-based mesh generation frameworks to quadrilateral mesh generation by introducing an additional vertex in the sequence representation.
2 The paper propose a topology-aware reinforcement learning fine-tuning method (tDPO) to enhance QuadMesh quality

**Strengths:**

1. Indeed, this paper extends existing triangle-based mesh generation methods to quadrilateral mesh generation.

2. The paper introduces a reinforcement learning strategy with one reward function that encourages long continuous edges and one penalty function that discourages fractures, aiming to improve mesh quality.

**Weaknesses:**

Although this paper is the first to extend autoregressive mesh generation to quadrilateral meshes, the extension—essentially adding one additional vertex—feels rather trivial and not strongly innovative.

The training process is also computationally expensive, requiring 64 A100 GPUs for 7 days, which makes it difficult for other researchers to reproduce the results. If the authors do not plan to release the code and weights, the paper’s academic contribution will be quite limited. Furthermore, the model is trained on proprietary licensed assets, which further reduces reproducibility and makes independent verification challenging.

Overall, this paper follows a typical data-driven approach—collect/label/clean a large dataset and train a large model. The methodology itself is not particularly innovative or inspiring, and it is unclear how future researchers could benefit from it.

**Questions:**

I could like the author to report NC and |NC| similar to other papers.

---

> ### Author Response · Authors · 2025-11-21
>
> We thank you for your review and appreciate the opportunity to clarify our contributions and address your concerns regarding novelty and reproducibility.
>
> #### **1. On Novelty and Contribution: A Scalable Path for Production-Ready 3D Generation**
>
> We respectfully disagree with the characterization of our work as a "trivial extension." Our work represents the **first successful and scalable framework for native quadrilateral mesh generation**, addressing the critical "last mile" problem of generating meshes that are not just viewable, but usable in professional workflows.
>
> *   **More Than "Adding a Vertex":** The shift to native quads introduces unique instability that simple extensions cannot handle. We solved this through a **holistic design**:
>     1.  **Scalable Architecture:** We deliberately chose a simple, padding-based representation over complex compressed tokenizers. As shown in **Appendix D.3**, this direct approach avoids the inductive biases of compression (e.g., MeshAnything, BPT), allowing the model to learn the precise global topology required for fracture-free quads.
>     2.  **Novel Training Strategy:** Naive training fails. Our **Curriculum Learning** (annealing `r`) and **tDPO** strategies were essential innovations to stabilize convergence and optimize topological rewards, which is impossible with triangle-only methods.
>     3.  **Bridging the Gap:** By proving that native generation is solvable and superior to post-hoc conversion, we provide a new direction for the community to move beyond "visual-only" generation.
>
> #### **2. On Reproducibility and Resource Accessibility**
>
> We acknowledge the high computational cost and are deeply committed to making our work accessible to the research community.
> *   **Open Source Plan:** We are actively working to release the **codebase**, **pre-trained model weights**, and a **substantial portion of our curated dataset** (including the benchmark).
> *   **Immediate Access:** To facilitate immediate reproducibility and testing, we will provide a public API and an online interactive demo. This ensures that researchers can evaluate and build upon our strong baseline without needing 64 A100s.
>
> #### **3. On Additional Topological Metrics (NC)**
>
> Thank you for this suggestion. We have computed the Normal Consistency (NC) metric to evaluate surface normal quality.
> The Normal Consistency (NC) measures the alignment of surface normals between the generated mesh $\mathcal{G}$ and the ground truth $\mathcal{P}$. It is defined as the average absolute dot product of the normals of nearest-neighbor points.
>
> As shown in the table below, QuadGPT achieves the highest NC score (**0.840**), significantly outperforming state-of-the-art baselines.
> As shown in the table below, QuadGPT achieves the highest NC score (**0.840**), significantly outperforming state-of-the-art baselines. This confirms that our method's superior topological structure also translates to higher geometric fidelity and surface smoothness.
>
> | Method | NC $\uparrow$ |
> | :--- | :---: |
> | MeshAnythingv2  | 0.667 |
> | BPT  | 0.737 |
> | DeepMesh | 0.534 |
> | FastMesh  | 0.718 |
> | **Ours** | **0.840** |

---

> ### Author Response · Authors · 2025-11-27
> **A Kind Reminder on the Author Rebuttal**
>
> Dear Reviewer,
>
> Thank you again for your time and insightful feedback. We have posted our author rebuttal, which includes detailed responses and several new experiments conducted to directly address the points raised in the reviews. We would be grateful if you could take a look at our response when you have a moment. We are happy to provide any further clarifications.

---

### Official Review · Reviewer_87rP · 2025-10-30

**Soundness:** 3
**Presentation:** 3
**Contribution:** 4
**Rating:** 8
**Confidence:** 4

**Summary:**

This paper presents an auto-regressive mesh generation model that can produce both triangular meshes and quadrilateral meshes. The model is trained with a multi-stage training strategy: 1) pre-training on triangular meshes only, 2) fine-tuning on a mixture of triangular and quadrilateral meshes, 3) reinforcement learning post-training to enhance the topology of the generated meshes.

**Strengths:**

1. This paper represents the first work on quadrilateral mesh generation. It proposes a hybrid representation for triangles and quadrilaterals, enabling the generation of artist-like 3D meshes input point clouds.

2. The paper presents a multi-stage end-to-end training framework, which incorporates both traditional next-token-prediction and sequence-level reinforcement learning supervision. The generation capability is improved in a targeted manner.

3. As observed in the metrics and qualitative study, the model achieves superior results. The paper also provides many potential applications of the generated meshes, highlighting its practical usefulness.

**Weaknesses:**

1. As the model supports both triangular and quadrilateral mesh generation, it would be better if treated triangular mesh generation as an evaluation task. It is interesting to study:
    - whether training on the mixed representation helps the generation of triangular meshes,
    - whether the proposed reinforcement learning also improves the generation of triangular meshes.

    It is also a natural request since previous works mainly focus on generating triangular meshes.

2. How well model follows the `quad-dominance parameter`? Now that we have mixed representations, *i.e.* triangular and mixed triangular and quadrilateral representations. Is the `quad-dominance parameter` sufficient for conditioning the model to generate the desired mesh representation? It would be better to show different generation results (triangular and mixed representations) for the same geometry.

**Questions:**

1. As one of the advantages of generating quadrilateral meshes is to save tokens (representing two triangles ~2 x 9 coords with one quadrilatero ~12 coords) It is also interesting to study how much the current model can represent complex geometries with less tokens compared to previous methods.

---

> ### Author Response · Authors · 2025-11-21
>
> We sincerely thank you for your positive assessment of our work, particularly for recognizing our contribution as "excellent" and the paper as a "good paper." We appreciate your insightful questions, which have helped us strengthen our evaluation.
>
> #### **1. On Evaluating Triangle Mesh Generation**
>
> We appreciate this valuable suggestion and have incorporated a rigorous comparison in **Section 4.4.3** between our model and a baseline (**TriGPT**) trained exclusively on triangular meshes.
> We emphasize that for geometric fidelity (CD/HD), evaluating on the converted quad mesh is equivalent to evaluating on the triangle mesh, but we advocate for a shift in focus to **topological quality**, as professional workflows prioritize the usability of quadrilateral meshes.
>
> *   **Impact of RL on Triangle Generation:**
>     Our experiments with TriGPT confirm that RL reduces fracture rates but faces fundamental limitations in improving **global topology** (edge flow). Topological rewards require quadrilateral structures, forcing a triangle-based model to rely on inaccurate post-processing conversion, which creates a significant bottleneck and motivated our pivot to **native quad generation**.
>
> *   **Impact of Mixed Training on Triangle Generation:**
>     Our results indicate that mixed training does not directly yield improvements in geometric fidelity for triangle generation compared to the pure triangle baseline (CD 0.065 vs 0.062). However, its true value lies in enabling the subsequent RL stage to bypass the conversion bottleneck, unlocking massive improvements in topological quality unattainable with a triangle-only pipeline.
>
> #### **2. On the Quad-Dominance Parameter `r`**
>
> We clarify that `r` functions as a **binary indicator** (0 or 1) to switch between mesh representations, rather than as a continuous controller for fine-grained ratio adjustment.
>
> *   **Curriculum Learning:** The primary role of `r` is to enable our **curriculum learning strategy**, where we anneal a sampling threshold from 0 to 1 during training. When a random probability $p < \text{threshold}$, we use the native quad representation (`r=1`); otherwise, we use the triangulated representation (`r=0`), allowing the model to master geometric structure before tackling complex topological rules.
> *   **Inference Behavior:** During inference, we set `r=1` to instruct the model to generate quad-dominant meshes, consistent with our training data. We avoid enforcing a rigid, user-defined quad ratio, as our model adaptively determines the optimal distribution (typically ~80%) to meet professional standards without overriding learned topological priors.
>
> #### **3. On Token Efficiency**
>
> Token efficiency is a key practical advantage of our approach, although our primary focus remains on quality rather than aggressive compression.
>
> *   **Quantitative Efficiency Analysis:** For a typical mesh with 10,000 faces and an 80% quad ratio (topologically equivalent to 18,000 triangles), our native representation uses 120,000 tokens ($10,000 \times 12$) versus 162,000 tokens ($18,000 \times 9$) for triangles. This results in a **~26% reduction** in sequence length (ratio $\approx 0.74$).
> *   **Architectural Design Philosophy:** While recent approaches such as MeshAnything or BPT achieve higher compression rates, we argue that such "over-design" can be counterproductive. As demonstrated in **Appendix D.3**, aggressive compression may accelerate early convergence but restricts the model's final performance ceiling. Native quad topology is inherently more challenging to learn than triangle topology, and introducing additional compression mechanisms would likely exacerbate training difficulties and degrade the final output quality. Consequently, we opted for a simple, direct representation to avoid harmful inductive biases, allowing the model to scale effectively and achieve superior topological quality.

---

> > ### Comment · Reviewer_87rP · 2025-11-25
> >
> > I appreciate the authors for addressing my concerns. I believe this paper holds significant practical value:
> >
> > - **Practicalness**. Unlike triangular meshes, which are relatively deterministic representations, native quad meshes encapsulate human design preferences. Consequently, leveraging data-driven learning directly from such preference-embedded data brings artistic mesh generation one step closer to practical deployment, endowing it with enhanced editability and flexibility.
> >
> > - **Data Pyramid**. This hybrid representation enables the model to undergo pre-training on large-scale triangular mesh datasets, allowing it to incorporate human design preferences through post-training refinement, which shows a similar concept to LLM methods.
> >
> > - **Scalability**. By discarding certain human heuristics inherent in mesh token compression and adopting a simpler mesh token ordering strategy, the approach aligns with the insights from "The Bitter Lesson" -- prioritizing scalable training paradigms. This positions the model to achieve a higher ceiling in data-driven generative performance.
> >
> > **Suggestions**. I would recommand that the authors incorporate part of the discussion from "D.3 Analysis of compressed vs. direct coordinate representation" (appendix) into the main body to better illustrate the rationale behind this simple design, but avoid use strong words like "over-designed".

---

> > > ### Author Response · Authors · 2025-11-27
> > >
> > > We sincerely appreciate your thoughtful review. Your recognition of our work's practical value and alignment with scalable methods is very encouraging.
> > > Following your excellent suggestion, we will integrate the core discussion on tokenization from the appendix into the main text and revise the language as you advised.
> > > Thank you again for your strong support.

---

### Author Response · Authors · 2025-12-02
**Summary for Area Chair**

Dear Area Chair,

We are truly grateful for your willingness to invest your precious time in conducting a comprehensive review of our submission and rebuttal, despite the unexpected circumstances.

To assist your evaluation, we provide a brief summary of our work.

**Paper Overview**
* This paper introduces **QuadGPT**, the first end-to-end autoregressive framework for native quadrilateral mesh generation. Compared with previous approaches that rely on triangle generation followed by heuristic conversion, QuadGPT employs a **unified padding-based tokenization** method and a specialized **tDPO** fine-tuning module, achieving superior geometric accuracy while optimizing global edge loops for professional usability. Our work addresses the "last mile" problem in 3D generation, significantly surpasses state-of-the-art baselines in both geometric fidelity and topological quality, effectively bridging the gap between generative AI models and **production-ready** 3D content creation.

**Initial Ratings**
8, 6, 4, 4(Avg: 5.5).

**Existing Discussions with Reviewers**
* **Reviewer 87rP :** "I appreciate the authors for addressing my concerns. I believe this paper holds significant practical value"

We regret that the other Reviewers (cNgg, SA17, N2bc) have not yet responded to our rebuttal. However, we have provided **detailed responses** and **conducted extensive new experiments** to fully resolve their initial concerns regarding data fairness and novelty.

**Recognized Strengths by All Reviewers**

* Pioneering end-to-end framework for native quadrilateral mesh generation.
* Impressive empirical results and superior topological quality.
* Clear presentation and easy to follow.

**Main Concerns and Questions Resolved:**

| Issue | Solution |
| :--- | :--- |
| **Data Fairness & Baselines** | Conducted two rigorous experiments to isolate algorithmic contribution: (1) **QuadGPT-OS** trained solely on open-source data (ShapeNet, Objaverse-XL) still outperforms baselines (Fig. 15, Appx D.4); (2) **TriGPT + Converter** baseline trained on the exact same dataset proves native generation yields superior topology (Table 3). |
| **Novelty & Contribution** | Clarified that the method is a holistic solution to unique instability problems in quad generation. Emphasized the synergy of **Systematic Design** choices: Padding-based tokenization for scaling, **curriculum learning** for stability, and **tDPO** for topological optimization. |
| **Reproducibility & Resources** | Addressed concerns about compute costs and proprietary data by committing to **open-source the codebase, pre-trained weights, and a curated benchmark dataset**. We have already launched a public website for low-resource verification. |
| **Tokenization Strategy** | Justified the choice of padding over compression in Appendix D.3. Demonstrated that while compression accelerates convergence, it introduces inductive biases that harm **topological precision**, whereas direct representation achieves a higher performance ceiling. |
| **Control Parameter **r****  | Clarified that **r** is a binary indicator for **curriculum learning** rather than a continuous ratio controller. Explained that the model adaptively determines the optimal quad distribution (~80%) for professional standards. |
| **Metrics & Comparisons** | Reported **Normal Consistency (NC)** metric (0.840) outperforming SOTA. Added discussion on **Blossom-Quad/Blender**, clarifying that native generation solves global planning problems that local conversion algorithms cannot handle. |

We would like to express our sincere gratitude to all reviewers for their constructive feedback. We are confident that our revisions, supplementary experiments, and comprehensive responses have effectively addressed every concern raised. We respectfully request that our rebuttal be taken into account in your decision-making process, and we again extend our deepest thanks for your dedicated efforts in advancing the review process amid these challenging times.

Yours sincerely, The Authors

---

### Meta-Review · Area_Chair_bHr3 · 2026-01-06

**Summary:**

This paper presents QuadGPT, an autoregressive framework for generating native quadrilateral and mixed triangle–quad meshes from point clouds, using a unified padding-based representation, an hourglass Transformer backbone, and a topology-aware RL finetuning stage. Reviewers agree that the problem is important, the results are strong, and that this is effectively the first serious system for direct quad mesh generation with production-style topology. The main reservations concern limited conceptual novelty beyond integrating known components, fairness of comparisons against triangle-based baselines due to data and capacity differences, and the high computational cost and reproducibility. Given the strength and breadth of the empirical results, and the additional experiments in the rebuttal, I lean toward recommending acceptance.

**Reviewer Concerns:**

The rebuttal convincingly addresses several key issues. Concerns about unfair comparisons and reliance on proprietary data are mitigated by added experiments with a model trained only on public data and a controlled triangle-only baseline followed by conversion, showing that the native quad formulation and training strategy themselves matter. Questions about the simplicity of the padding-based representation, and about the quad-dominance parameter, are clarified: padding is justified as a deliberate design choice for topology fidelity and weight reuse, and the parameter is framed as a binary curriculum signal rather than a precise control knob. Reproducibility is partly addressed by explicit plans to release code, weights, and a substantial data subset, and by adding standard metrics. Remaining reservations are mainly about limited algorithmic novelty and the high computational cost, which the rebuttal cannot fully remove but which do not undermine the technical soundness or practical value.

**Reviewer Scores:**

Reviewer 87rP was strongly positive and became more so after the rebuttal; their judgment would remain clearly in favor of acceptance. Reviewer N2bc was already mildly positive, and the added experiments directly speaking to their questions would likely reinforce that stance. Reviewers cNgg and SA17 started slightly below the acceptance threshold, largely because of novelty and fairness concerns; with the new baselines, ablations, and open-source commitments, their positions would likely move toward a more neutral or mildly positive view.

---

### Decision · Program_Chairs · 2026-01-26

Accept (Poster)